# Hot and heavy dark matter from a weak scale phase transition

Iason Baldes[1*], Yann Gouttenoire[2] and Filippo Sala[3]

**1** Service de Physique Théorique, Université Libre de Bruxelles,
Boulevard du Triomphe, CP225, 1050 Brussels, Belgium
**2** School of Physics and Astronomy, Tel-Aviv University, Tel-Aviv 69978, Israel
**3** Laboratoire de Physique Théorique et Hautes Énergies, CNRS,
Sorbonne Université, Paris, France

* iasonbaldes@gmail.com

## Abstract

We point out that dark matter which is produced non-adiabatically in a phase transition (PT) with fast bubble walls receives a boost in velocity which leads to long free-streaming lengths. We find that this could be observed via the suppressed matter power spectrum for dark matter masses around $10^8 - 10^9$ GeV and energy scales of the PT around $10^2 - 10^3$ GeV. The PT should take place at the border of the supercooled regime, i.e. approximately when the Universe becomes vacuum dominated. This work offers novel physics goals for galaxy surveys, Lyman-$\alpha$, stellar stream, lensing, and 21-cm observations, and connects these to the gravitational waves from such phase transitions, and more speculatively to possible telescope signals of heavy dark matter decay.

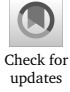 Check for updates

# 1 Introduction

Two major constraints on the properties of dark matter (DM) come from observations of the CMB and the large scale structure of matter. The former is a powerful probe of the energy content of the Universe, precisely constraining the matter content of baryons and dark matter, along with the other $\Lambda$CDM parameters [1]. Observations of the matter power spectrum, on the other hand, while helping pin down the DM density, also provide strong constraints on the DM velocity dispersion.

The matter power spectrum has been measured at large scales through galaxy surveys [2], at intermediate scales through weak lensing observations [3], and at the smallest scales through Lyman-$\alpha$ forest data [4–7], Milky Way satellite [8], stellar stream [9], and strong lensing observations [10–13]. The reported limits on a small scale cut in the spectrum are typically given in the context of standard warm DM, i.e. two component fermionic DM that freezes out while relativistic, and are in the range $m_{\text{WDM}} \gtrsim (2-7)$ keV [4–13]. Future observations of the 21-cm signal could push this constraint to $m_{\text{WDM}} \gtrsim 15$ keV [14]. The $m_{\text{WDM}}$ limit is not applicable model independently. When considering alternative models, one can instead calculate and compare with the free streaming length or velocity dispersion. Taking a fiducial value, $m_{\text{WDM}} \gtrsim 5$ keV, corresponds to DM free streaming length at matter-radiation equality of $\lambda(t_{\text{eq}}) \approx 0.1$ Mpc, or a mean velocity [15–17]

$$v(t_{\text{eq}}) = \left( \frac{4}{11} \frac{94\,\text{eV}}{m_{\text{WDM}}} \Omega_{\text{DM}} h^2 \right)^{1/3} \frac{3.15\, T_\gamma^{\text{eq}}}{m_{\text{WDM}}} \simeq 5 \times 10^{-5} \left( \frac{5\,\text{keV}}{m_{\text{WDM}}} \right)^{4/3}, \tag{1}$$

where $T_\gamma^{\text{eq}} \simeq 0.8$ eV is the photon temperature at matter-radiation equality. It is possible to have DM with a non-negligible $v(t_{\text{eq}})$, which we will generically refer to as non-cold DM (NCDM), with a mass much larger than 5 keV. Known examples are DM coming from the evaporation of priomordial black holes [18–21], freeze-in [22–24], decay of heavier particles [22, 24, 25], or with large interactions [26, 27]. The effect of early phase transitions (PTs) directly on the late time DM velocity, however, through the kick the DM particles receive at the phase boundary, has so far not been considered. For alternative mechanisms where PTs modify the matter power spectrum, but at (much) lower temperatures, see [28–36].

Particles which gain a mass when crossing the bubble wall separating the high and low temperature phases, also obtain a boost in the original plasma (eventual CMB) frame [37]. In

principle, if the DM interactions with the thermal bath following the PT are sufficiently weak, the DM will not return to kinetic equilibrium. In this way, the momentum gained at the time of the PT, suitably redshifted, can lead to NCDM. In the case of DM simply gaining a mass during the PT, such as in [38–40], however, the velocity dispersion is negligible compared to current limit even if the PT is supercooled. The reason is the presence of irreducible interactions with the scalar driving the PT, which means the DM will not retain a large enough velocity to approach free streaming limits. Similar conclusions hold in models of supercooled composite DM; although initially highly boosted, theoretically unavoidable interactions lead to deep-inelastic scatterings of the DM with the dilaton field following the PT, which would also bring the DM back into kinetic equilibrium in this case [41, 42].

We therefore consider the DM production scenario introduced by Azatov, Vanvlasselaer, and Yin; during a PT in which a scalar gains a VEV $v_\phi$, DM with mass $m_{\rm DM} \gg v_\phi$ is produced non-adiabatically across the bubble wall [43–45]. The DM is produced with a large Lorentz factor in the original plasma frame, which when redshifted leads to a non-negligible $v(t_{\rm eq})$. The crucial qualitative difference, in this case, is that $m_{\rm DM}$ may be super-heavy, with mass sufficiently above the temperature of the bath following the PT, so that interactions with the scalar $\phi$ are out-of-equilibrium. Note in this scenario, we must assume a reheating temperature after the usual cosmological inflation, $T \ll m_{\rm DM}$, so that the DM begins with effectively zero abundance in the initial radiation dominated phase, as we want the majority of our DM to be produced with a kick during the PT. Similarly the inflaton should not decay significantly into DM particles. (Alternatively, we may imagine some non-standard expansion history which dilutes DM prior to the epoch of the PT.)

Finally, we remind the reader, that in this NCDM picture $N_{\rm eff}$ limits at BBN are weaker than limits from structures because DM is far less abundant at BBN times compared to a standard hot thermal relic.

## 2   Phase Transition

We consider a scalar field $\phi$, real or complex, which gains a VEV $v_\phi$ during an early Universe PT. We assume an initially radiation dominated Universe following standard cosmological inflation. Bubbles nucleate at some temperature $T_n$, expand, collide, and convert the Universe to the new phase. Two qualitatively different expansion histories present themselves as possibilities. If bubbles nucleate early enough, the Universe remains radiation dominated throughout this epoch. If instead, nucleation is delayed, the radiation density may drop below the false vacuum density and the Universe enters an additional inflationary phase at temperature defined by

$$\frac{g_* \pi^2}{30} T_{\rm infl}^4 \equiv \Lambda_{\rm vac} \equiv c_{\rm vac} v_\phi^4. \tag{2}$$

Here $g_*$ are the effective radiation degrees-of-freedom and $c_{\rm vac}$ is a dimensionless, model dependent, number parametrizing the vacuum energy difference. For brevity and simplicity, we assume rapid scalar condensate decay following the PT, see App. A for discussion on how this can be realised. The temperature of the radiation bath just after the PT is therefore given by

$$T_{\rm RH} \simeq {\rm Max}[T_n, T_{\rm infl}]. \tag{3}$$

The leading order pressure from the change in particle masses across the bubble wall, in the ultra-relativistic ballistic regime, is given by [37, 46]

$$\mathcal{P}_{\rm LO} \simeq \sum_a \Delta(m_a^2) \int \frac{d^3 p f_a^{\rm eq}}{(2\pi)^3 2E_a} \equiv g_a \frac{v_\phi^2 T_n^2}{24}, \tag{4}$$

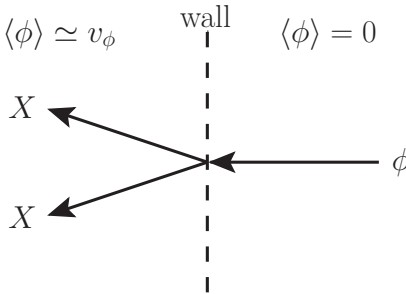

Figure 1: When light $\phi$ quanta enter the bubble of new phase, they can produce $X + X$ DM pairs, which are highly boosted in the original plasma frame.

where $\Delta(m_a^2)$ denotes the mass squared difference between the two phases, $f_a^{\text{eq}}$ is the equilibrium number density in the symmetric phase, and $g_a$ is a convenient parametrization of the effective degress-of-freedom gaining a mass of order $v_\phi$. For sufficiently small $T_n$, one has $\mathcal{P}_{\text{LO}} < \Lambda_{\text{vac}}$, and an effectively run-away wall. In this case, the Lorentz factor of the wall grows linearly with distance and at collision is $\gamma_{\text{wp}} \simeq R_{\text{col}}/(3R_n)$ [47], where $R_n$ and $R_{\text{col}}$ are the bubble radii at nucleation and collision respectively. The bubbles nucleate with a typical size $R_n \equiv A_{\text{bub}}/T_n$ with $A_{\text{bub}} \sim 1 - 10$. At collision, $R_{\text{col}} \simeq (8\pi)^{1/3} v_w/(\beta_H H)$ where $\beta_H$ is the inverse timescale of the transition normalised to Hubble, $H \propto T_{\text{RH}}^2/M_{\text{Pl}}$, where we define $M_{\text{Pl}}$ as the reduced Planck mass, and $v_w \simeq 1$ is the wall velocity. Typical values for supercooled PTs are $\beta_H \sim 10$. Close to bubble collision, when the majority of the volume is being converted to the true vacuum, the bubble wall Lorentz factor as measured in the plasma frame is therefore given by

$$\gamma_{\text{wp}} \simeq \frac{2\sqrt{10} T_n M_{\text{Pl}}}{\pi^{2/3} A_{\text{bub}} \beta_H g_*^{1/2} T_{\text{RH}}^2} \,. \tag{5}$$

The emission of soft quanta with phase dependent masses induces additional pressure [46,47]. If no gauge boson obtains a mass at the PT, then the resulting pressure is subleading with respect to the LO one of Eq. (4), and Eq. (5) for the Lorentz factor is valid. We limit our discussion to this case in the rest of the paper.

## 3 Non-Adiabatically Produced DM

We now introduce a real scalar DM candidate, with non-negligible DM mass in the symmetric phase, together with an interaction with the scalar field gaining a VEV

$$\mathcal{L} \supset -\frac{1}{2} m_{\text{DM}}^2 X^2 - \frac{1}{4} \lambda \phi^2 X^2 \,. \tag{6}$$

For concreteness, we phrase our discussion assuming $\phi$ is a real scalar. To avoid problems with domain walls when $\phi$ gains a VEV, the symmetry $\phi \rightarrow -\phi$ should be explicitly broken by other terms, that can be kept small enough to not influence the rest of this paper. Our findings will also largely be valid for a complex $\phi$, as we will comment on later.

We assume zero initial DM abundance in the symmetric phase. This requires negligible production via inflaton decay, and a Boltzmann suppression of thermal processes which would generate a DM population following standard cosmological inflation, which can be achieved provided $m_{\text{DM}}/T$ is always large enough, e.g. $m_{\text{DM}}/T \gtrsim \mathcal{O}(30)$ to remain under the observed abundance via freeze-in. Alternatively, there may be some additional dilution mechanism in play at high $T$. Then the dominant DM relic abundance may be produced non-adiabatically

when light $\phi$ quanta enter the bubbles, as we consider here. The probability of DM pair production reads [43–45][1]

$$P(\phi \to X + X) = \frac{\lambda^2 v_\phi^2}{192\pi^2 m_{\text{DM}}^2} , \tag{7}$$

assuming the Lorentz factor, introduced in Eq. (5), satisfies

$$\gamma_{\text{wp}} \gtrsim \frac{L_w m_{\text{DM}}^2}{T_n} \approx \frac{m_{\text{DM}}^2}{\sqrt{c_{\text{vac}}} v_\phi T_n} , \tag{8}$$

where we have approximated the wall width as the inverse of the scalar mass $L_w \approx 1/m_\phi \approx 1/(\sqrt{c_{\text{vac}}} v_\phi)$ (see e.g. [41, 47]). The above is known as the anti-adiabatic regime, for smaller $\gamma_{\text{wp}}$ there is a further sharp suppression of the production probability. The DM abundance normalised to entropy, in the anti-adiabatic regime, is then given by

$$Y_{\text{DM}} = \frac{45\zeta(3)}{2\pi^4 g_{*s}} \frac{\lambda^2 v_\phi^2}{96\pi^2 m_{\text{DM}}^2} \left(\frac{T_n}{T_{\text{RH}}}\right)^3 , \tag{9}$$

where $g_{*s}$ are the entropic degrees of freedom. Here the first factor represents the number density of $\phi$ quanta normalized to entropy (we have assumed an approximately massless $\phi$ in the symmetric phase), the second is the $X + X$ production probability multiplied by two as the DM is being pair produced, and the third is an entropy dilution factor. In general, there are up to two choices of $T_{\text{RH}}$ which will match the observed value, $Y_{\text{DM}} m_{\text{DM}} = 0.43$ eV. One corresponds to the PT occuring in the radiation dominated regime, $T_n > T_{\text{infl}}$, and the other in the supercooled vacuum dominated regime, $T_n < T_{\text{infl}}$.

Note that, as first worked out in [43], pair production induces only a small additional contribution to the pressure, Eq. (4), approximately given by $g_a \to g_a + \lambda^2 \log(1 + \gamma_{\text{wp}} T m_\phi / m_{\text{DM}}^2)/(32\pi^2)$ which leaves our estimate of the bulk bubble properties during expansion effectively unchanged. Locally, the momentum exchange will distort the wall, although to what extent this would, e.g., modify the effective wall tension is an open question.[2] (We do not attempt to solve scalar equations of motion in the presence of DM pair production in the current work.)

## 4  Non-Cold Heavy DM

We must also determine $v(t_{\text{eq}})$. Consider the kinematics of light quanta entering the bubble and pair producing DM. The situation is illustrated in Fig. 1. Going into the time independent wall frame, which will allow us to use energy conservation across the wall, an incoming $\phi$ quantum has energy $E \sim \gamma_{\text{wp}} T_n$. To gain intuition, consider the special case in which the outgoing $X$ quanta share the incoming energy equally. Then, in the wall frame, the DM Lorentz factor is $\gamma_{\text{xw}} \sim \gamma_{\text{wp}} T_n / 2m_{\text{DM}}$. It is a good and conservative approximation to ignore the momentum transverse to the direction of the wall velocity. Then the DM Lorentz factor in the plasma frame is $\gamma_{\text{xp}} \sim \gamma_{\text{wp}}/2\gamma_{\text{xw}} \approx m_{\text{DM}}/T_n$. (A more precise derivation is given in App. B.) The initial DM momentum is therefore $p_{\text{DM}}(T_{\text{RH}}) \simeq m_{\text{DM}}^2/T_n$. Accordingly, the redshifted velocity at matter-radiation equality is given by

$$v(t_{\text{eq}}) \simeq \left(\frac{g_{*s}(T_{\text{eq}})}{g_{*s}(T_{\text{RH}})}\right)^{1/3} \frac{T_\gamma^{\text{eq}} m_{\text{DM}}}{T_{\text{RH}} T_n} , \tag{10}$$

---

[1]Taking into account the different normalizations of the coupling, we find a factor of two smaller production probability than [45, Eq. (58)].

[2]We thank the referee for pointing this out.



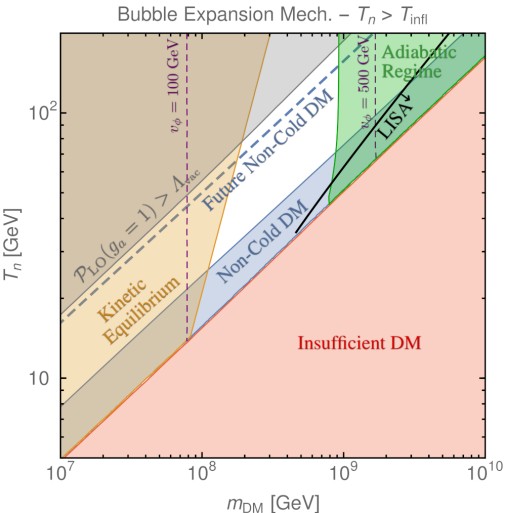
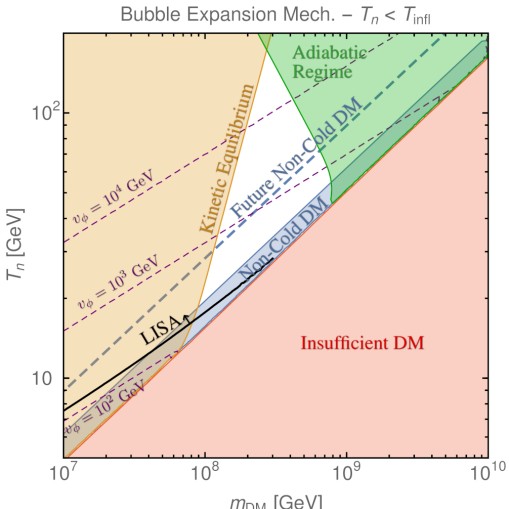

Figure 2: Heavy non-cold DM from fast bubble walls in the plane of nucleation temperature $T_n$ vs DM mass $m_{\mathrm{DM}}$, for $T_n > T_{\mathrm{infl}}$ (left) and $T_n < T_{\mathrm{infl}}$ (right). We set $\lambda = 1$ and $c_{\mathrm{vac}} = 10^{-2}$. Viable non-cold DM can be produced in the range $m_{\mathrm{DM}} \approx 10^8 - 10^9$ GeV (white area), delimited by: the requirement of the anti-adiabatic regime, Eq. (8), at bubble collision with $A_{\mathrm{bub}} = 3$ and $\beta_H = 10$ (green); too small DM yield, Eq. (9), even with the DM number maximizing choice $T_{\mathrm{infl}} = T_n$ (red); kinetic equilibration, i.e. violation of Eq. (11) (tan); the bubbles not running away, i.e. $\Lambda_{\mathrm{vac}} < \mathcal{P}_{\mathrm{LO}}$ of Eq. (4) (gray, left); the warm DM velocity limit for $m_{\mathrm{WDM}} \gtrsim 5$ keV, corresponding to $v(t_{\mathrm{eq}}) \lesssim 5 \times 10^{-5}$ (blue). The blue NCDM region spans $v_\phi/T_n \approx 5-7$ (left) and $v_\phi/T_n \approx 7-12$ (right). The dashed blue line shows the future sensitivity at $m_{\mathrm{WDM}} = 15$ keV, $v(t_{\mathrm{eq}}) \approx 10^{-5}$. Purple dashed contours show the VEV, $v_\phi$. The region below (above) the black contour on the left (right) panel can be tested by LISA with a signal-to-noise ratio SNR > 5. In the left panel, however, this lies outside the valid domain of parameter space for the DM model.

where $g_{*s}(T_{\mathrm{eq}}) \simeq 3.91$.

Finally, we need to ensure that scatterings with the thermal bath, namely $X + \phi \to X + \phi$ interactions, do not spoil our estimate of the final DM velocity. The strictest condition comes from the four-point vertex in Eq. (6). A simple criterion is found by demanding the scattering rate, weighted by the fractional momentum loss, be below Hubble for a point in parameter space to be considered viable

$$n_\phi \sigma(X\phi \to X\phi) v_{\mathrm{Møl}} \frac{\delta p_{\mathrm{DM}}}{p_{\mathrm{DM}}} = n_\phi \frac{\lambda^2 p_{\mathrm{CM}}}{8\pi \hat{s}^{3/2}} < H, \tag{11}$$

where $\sqrt{\hat{s}}$ is the centre-of-mass energy, $p_{\mathrm{CM}}$ is the centre-of-mass momentum, and for this interaction $\delta p_{\mathrm{DM}} \approx p_{\mathrm{DM}}/2$ (the above formulation does not hold for $t$-channel scatterings, more on this below). In the relativistic regime, $p_{\mathrm{DM}} \approx m_{\mathrm{DM}}^2 T/(T_{\mathrm{RH}} T_n)$, $\hat{s} \approx 4 m_{\mathrm{DM}}^2 T^2/(T_{\mathrm{RH}} T_n)$, $p_{\mathrm{CM}} \simeq \sqrt{\hat{s}}/2$, and the number density of $\phi$ in the thermal bath is given by

$$n_\phi = \frac{g_\phi \zeta(3)}{\pi^2} T^3. \tag{12}$$

In terms of the temperature, the LHS of Eq. (11) scales as $T$, while the RHS scales as $T^2$. One may therefore worry that the condition will become increasingly more stringent for lower $T$. However, the above assumes massless $m_\phi$; for $T \lesssim m_\phi$ the number density $n_\phi$ quickly

becomes Boltzmann suppressed. Here we will make the assumption $m_\phi \sim \sqrt{c_{\mathrm{vac}}} v_\phi \sim T_{\mathrm{RH}}$ in the broken phase, typical for supercooled PTs, and evaluate the above condition at $T = T_{\mathrm{RH}}$. (The effective $m_\phi$ in the symmetric phase may be somewhat different, for example $m_\phi \sim T_n$ if it is dominated by thermal contributions to the effective potential at the time of the PT.)

There are also additional interactions with SM bath particles and $\phi$ quanta, involving soft $t$-channel scalar exchange, which we have carefully checked do not lead to a significant reduction in the $X$ momentum. The results are given in App. C. The conclusion of our detailed calculations, given therein, is that we are safe from a return to kinetic equilibrium provided inequality (11) holds. We also show that even if $m_\phi \ll T_{\mathrm{RH}}$, viable parameter space still exists, due to the scaling of $p_{\mathrm{CM}}$ and $\hat{s}$ at lower temperatures. Furthermore, if we instead considered a complex scalar $\phi = \rho e^{ia/v_\phi}$, then $X$ scatterings with the axion-like particle $a$ would be dominated by hard $t$-channel exchange of the radial mode. We show that these do not impact the estimate in Eq. (11) as long as $T_{\mathrm{RH}} \lesssim 10 T_n$.

Finally note, that in the parameter space of interest, the DM is always chemically decoupled following the PT, i.e. the annihilation rate $X + X \to \phi + \phi$ is also below $H$. Elastic self-interactions between the DM can reduce the $m_{\mathrm{WDM}}$ constraint by $\sim 20\%$ [48, 49], however, because of the super-heavy nature of our DM, its non-gravitational self-interactions are also completely negligible.

We now combine all our calculations and constraints and display the results in Fig. 2. As summarized in the figures, we see NCDM is possible with this mechanism at masses $m_{\mathrm{DM}} \sim (10^8 - 10^9)$ GeV. The NCDM is realized for nucleation temperatures $T_n \sim 10$ GeV, and reheating temperatures $T_{\mathrm{RH}} \sim (10 - 10^2)$ GeV. The underlying scale of the beyond the standard model (BSM) sector is $v_\phi \sim (10^2 - 10^3)$ GeV. The region close to the NCDM constraint could be tested by future observations targeting a cut in matter power spectrum at small scales. The bubble collisions following the PT will also result in a strong gravitational wave (GW) signal, which we turn to next.

## 5 Gravitational Wave Signal

We now detail the expected GW signal. For our mechanism, we require the bubbles to effectively run-away until collision, so that the majority of the vacuum energy is transferred to the walls. Accordingly, in giving an estimate of the expected GWs, it is appropriate to use the numerical results from Cutting et al. [50],

$$h^2 \Omega_{\mathrm{GW}}(f) \equiv h^2 \frac{d\Omega_{\mathrm{GW}}}{d\log(f)} = 2.0 \times 10^{-6} \times \left(\frac{\alpha}{1+\alpha}\right)^2 \frac{S_\phi(f)}{g_*^{1/3} \beta_H^2}, \tag{13}$$

where $\alpha$ is the energy released as bulk motion during the transition (which we approximate as the false vacuum energy) normalized to the radiation density. Here, the shape of the spectrum is governed by

$$S_\phi(f) = \frac{(a+b)\tilde{f}^b f^a}{b\tilde{f}^{(a+b)} + a f^{(a+b)}}, \tag{14}$$

where for PTs of our type the central numerical results indicate $a = 0.742$ and $b = 2.16$ [50]. (Also see [51–55].) The peak frequency of the signal today is

$$\tilde{f} = 15\,\mu\mathrm{Hz} \times \beta_H\, g_*^{1/6} \left(\frac{T_{\mathrm{RH}}}{10^3\,\mathrm{GeV}}\right). \tag{15}$$

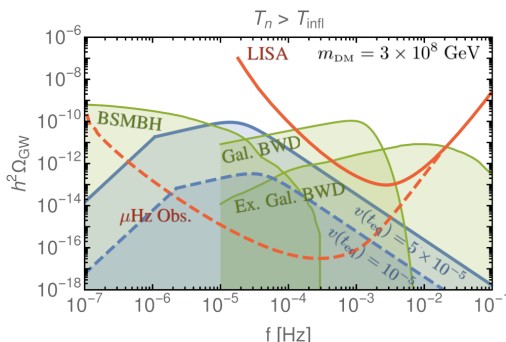
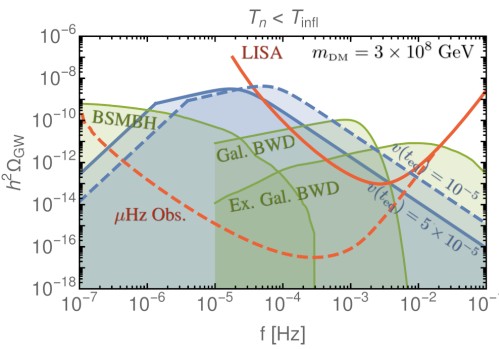

Figure 3: The solid (dashed) blue lines show the predicted gravitational wave spectrum for the PT corresponding to a DM mass $m_{\mathrm{DM}} = 3 \times 10^8$ GeV, $c_{\mathrm{vac}} = 10^{-2}$, $\lambda = 1$, and $v(t_{\mathrm{eq}})$ at the current limit of $m_{\mathrm{WDM}} = 5$ keV (future limit of $m_{\mathrm{WDM}} = 15$ keV). The former (latter) corresponds to a PT with $T_{\mathrm{RH}} \approx 40$ GeV ($T_{\mathrm{RH}} \approx 90$ GeV), in the case $T_n > T_{\mathrm{infl}}$, and with $T_{\mathrm{RH}} \approx 50$ GeV ($T_{\mathrm{RH}} \approx 150$ GeV), in the case $T_n < T_{\mathrm{infl}}$. In both cases lower (higher) DM masses would correspond to lower (higher) reheating temperatures and lower (higher) peak frequencies. We have assumed $\beta_H = 10$. The spectra are compared with power law integrated sensitivity curves, with signal-to-noise ratio SNR= 5, for LISA [61] and a future $\mu$Hz interferometer [62]. Estimated astrophysical foregrounds from binary super-massive black holes [63], galactic white-dwarf binaries [64] and extragalactic white-dwarf binaries [65] are also shown. Gravity gradient noise from asteroids (not shown) could also be significant up to $\sim 10^{-6}$ Hz [66]. The signal for the $T_n > T_{\mathrm{infl}}$ regime is below LISA expectations.

Finally, one should impose the correct $\Omega_{\mathrm{GW}} \propto f^3$ scaling for the initially super-horizon IR modes [56–60], corresponding to frequencies today below

$$
f_* = \left( \frac{a(T_{\mathrm{RH}})}{a(T_{\mathrm{today}})} \right) \times \frac{H(T_{\mathrm{RH}})}{2\pi} = 12\,\mu\mathrm{Hz} \times g_*^{1/6} \left( \frac{T_{\mathrm{RH}}}{10^3\,\mathrm{GeV}} \right).
$$

Now we are ready to use this spectrum together with our results for the NCDM. Accordingly, we take the prediction of $T_{\mathrm{RH}}$ for a given $m_{\mathrm{DM}}$ and $v(t_{\mathrm{eq}})$ and consider the estimated GW signal. The resulting spectra for two parameter points are shown in Fig. 3. We also calculated the SNR for LISA, strictly using the method given in [42], and display the contours which delineate SNR $= 5$ in Fig. 2. For $T_n > T_{\mathrm{infl}}$, the signal is suppressed by the scaling $\Omega_{\mathrm{GW}} \propto \alpha^2 \propto (T_{\mathrm{infl}}/T_n)^8$, as $\alpha \lesssim \mathcal{O}(1)$. Thus this regime can only be extensively probed through its induced small scale structure suppression, assuming $g_a \gtrsim 1$, and partly through far future GW observations. For $T_n \lesssim T_{\mathrm{infl}}$, instead, the amplitude of the GW signal is large. For lower values of $T_{\mathrm{RH}}$, however, the peak frequency is the IR of the LISA sensitivity. This qualitatively explains the behaviour of the SNR contours in Fig. 2. Note the entire allowed area for $T_n \lesssim T_{\mathrm{infl}}$, given our estimates, can be probed by LISA (even beyond the future NCDM region).

## 6 Conclusion

We investigated the possibility of dark matter being both heavy and non-cold as a result of a phase transition. In order to achieve sufficient high DM velocities at late times to be relevant for Lyman-$\alpha$ observations, we considered the non-adiabatic pair production mechanism first introduced in [43, 44]. We find viable non-cold DM compatible with Lyman-$\alpha$ bound in the mass range $m_{\mathrm{DM}} \sim (0.1-1)(M_{\mathrm{pl}}^2 T_\gamma^{\mathrm{eq}})^{1/3} \sim (10^8 - 10^9)$ GeV, with an underlying scale of the PT

$v_\phi \sim (10^2 - 10^3)(M_{\rm pl} T_\gamma^{\rm eq\,2})^{1/3} \sim (10^2 - 10^3)$ GeV, reheating temperature $T_{\rm RH} \sim (0.1-1) v_\phi$, and nucleation temperature $T_n \sim (0.1-1) T_{\rm RH}$. Despite the low $T_{\rm RH}$, which can provide a challenge due to washout, it may be possible to use the same PT (and mechanism) for baryogenesis [67–70].

The scale of the phase transition $v_\phi$ is intriguingly close to the electroweak scale. Our PT cannot naively be the EW one, even if some BSM physics made the latter first order, because weak gauge bosons getting a mass would prevent the bubble walls from running away and reaching the velocities of Eq. (5) [46,47], which are crucial for our mechanism. One may still speculate that the kind of PT discussed in this paper arises from the breaking of some global symmetry, which is tied to the mechanism of generation of the EW scale, as it could happen in composite models [71,72] or in supersymmetry [73–76]. We do not speculate further in this direction in this paper, we just provide further details on the coincidence of scales in App. D.

The rather unique signature of the heavy DM picture we presented is the combination of i) a suppression of structure at small scales, which will be interesting to precisely determine in future work, and ii) a large amplitude stochastic background of GWs [17,50,53,77–79] from the PT, with peak frequency in the range $f \sim (10^{-6} - 10^{-4})$ Hz.

Concerning other DM signals, direct detection is unfortunately beyond reach of conceivable future facilities. Coming to indirect detection, the number densities and hence annihilation signals are very small and, with the minimal content above, the DM is stable. If the $Z_2$ symmetry $X \to -X$ is broken, then the DM may decay and give a signal at high-energy telescopes.

# Acknowledgements

We are grateful to Miguel Vanvlasselaer for helpful correspondence.

**Funding information.** IB is a postdoctoral researcher of the F.R.S.–FNRS with the project '*Exploring new facets of DM.*' YG is grateful to the Azrieli Foundation for the award of an Azrieli Fellowship. FS acknowledges funding support from the Initiative Physique des Infinis (IPI), a research training program of the Idex SUPER at Sorbonne Université. We are grateful to GGI for hospitality and partial support during the completion of this work.

# A   Scalar Decay Rate

In the main text we have assumed the $\phi$ particles and/or condensate decays rapidly following the PT. Perhaps the simplest way this can be achieved, is by introducing a portal interaction to the SM Higgs. To illustrate this consider the interactions between the EW Higgs doublet $H$ and a real scalar $\varphi$,

$$\mathcal{L} \supset -\mu_h^2 |H|^2 - \lambda_h |H|^4 - \frac{\mu_\phi^2}{2} \varphi^2 - \frac{\lambda_\phi}{4} \varphi^4 - \frac{\lambda_{h\phi}}{2} \varphi^2 |H|^2, \tag{A.1}$$

where $\lambda_h \simeq 0.13$ is the EW Higgs self-quartic, and $\lambda_\phi \sim c_{\rm vac}$ is the exotic scalar analogue. In principle other terms are also allowed, however, the above will be sufficient to illustrate the idea. The minimum of the potential lies at $(v_\phi, v_{\rm EW})$ where $v_{\rm EW} \simeq 246$ GeV is the EW VEV and

$$\mu_h^2 = -\lambda_h v_{\rm EW}^2 - \frac{1}{2} \lambda_{h\phi} v_\phi^2, \tag{A.2}$$

$$\mu_\phi^2 = -\lambda_\phi v_\phi^2 - \frac{1}{2} \lambda_{h\phi} v_{\rm EW}^2. \tag{A.3}$$

Around the minimum, ignoring Goldstone directions, we introduce the massive scalar excitations $H = (v_{\text{EW}} + \tilde{h})/\sqrt{2}$ and $\varphi = v_\phi + \tilde{\phi}$. The physical mass eigenstates are

$$
\begin{pmatrix} h \\ \phi \end{pmatrix} = \begin{pmatrix} \cos\theta_{h\phi} & \sin\theta_{h\phi} \\ -\sin\theta_{h\phi} & \cos\theta_{h\phi} \end{pmatrix} \begin{pmatrix} \tilde{h} \\ \tilde{\phi} \end{pmatrix},
\tag{A.4}
$$

with associated mass eigenvalues

$$
m_h^2 = 2\lambda_h v_{\text{EW}}^2 \cos^2\theta_{h\phi} + 2\lambda_\phi v_\phi^2 \sin^2\theta_{h\phi} - \lambda_{h\phi} v_\phi v_{\text{EW}} \sin 2\theta_{h\phi},
\tag{A.5}
$$

$$
m_\phi^2 = 2\lambda_h v_{\text{EW}}^2 \sin^2\theta_{h\phi} + 2\lambda_\phi v_\phi^2 \cos^2\theta_{h\phi} + \lambda_{h\phi} v_\phi v_{\text{EW}} \sin 2\theta_{h\phi}.
\tag{A.6}
$$

We have introduced the usual mixing angle $\theta_{h\phi}$ between the two scalars, present once both have gained a VEV, which is given by

$$
\tan 2\theta_{h\phi} = \frac{\lambda_{h\phi} v_\phi v_{\text{EW}}}{\lambda_\phi v_\phi^2 - \lambda_h v_{\text{EW}}^2} \simeq \frac{2\lambda_{h\phi} v_\phi v_{\text{EW}}}{m_\phi^2 - m_h^2} \simeq
\begin{cases}
\dfrac{\lambda_{h\phi} v_\phi}{\lambda_h v_{\text{EW}}}, & \text{for } m_\phi < m_h, \\[2ex]
\dfrac{\lambda_{h\phi} v_{\text{EW}}}{\lambda_\phi v_\phi}, & \text{for } m_\phi > m_h.
\end{cases}
\tag{A.7}
$$

## A.1 Heavy $m_\phi$

Consider first the regime $m_\phi \gtrsim 2m_h \approx 250$ GeV. As $T_{\text{RH}} \sim m_\phi$, we assume the decay occurs in the unbroken electroweak (EW) phase. Demanding the decay rate into the SM Higgs doublet,

$$
\Gamma_{\phi \to HH} \simeq \frac{\lambda_{h\phi}^2 v_\phi^2}{8\pi m_\phi},
\tag{A.8}
$$

be above Hubble, translates into a condition

$$
\lambda_{h\phi} \gtrsim 10^{-7} \left(\frac{g_*}{100}\right)^{1/4} \left(\frac{10\, T_{\text{RH}}}{v_\phi}\right) \left(\frac{m_\phi}{10^4\, \text{GeV}}\right)^{1/2}.
\tag{A.9}
$$

Once the symmetries are broken, we therefore have

$$
\theta_{h\phi} \gtrsim 10^{-8} \left(\frac{g_*}{100}\right)^{1/4} \left(\frac{10\, T_{\text{RH}}}{v_\phi}\right) \left(\frac{m_\phi}{10^4\, \text{GeV}}\right)^{1/2} \left(\frac{10^{-2}}{\lambda_\phi}\right) \left(\frac{10^5\, \text{GeV}}{v_\phi}\right),
\tag{A.10}
$$

in the heavy $m_\phi$ regime.

## A.2 Light $m_\phi$

If, instead, $m_\phi$ is around or below the EW scale, the decay to SM Higgs bosons is kinematically disallowed, and the decay occurs in the broken EW phase. Through the mixing angle the $\phi$ can decay to SM fermions. In the $\theta_{h\phi} \ll 1$ limit, the rate is given by

$$
\Gamma_{\phi \to \bar{f}f} \approx \frac{N_c m_f^2 \theta_{h\phi}^2 m_\phi}{8\pi v_{\text{EW}}^2},
\tag{A.11}
$$

where $m_f$ is the fermion mass, and $N_c$ are the number of colours. The decay rate is faster than Hubble provided

$$
\theta_{h\phi} \gtrsim 10^{-6} \left(\frac{g_*}{100}\right)^{1/4} \left(\frac{3}{N_c}\right)^{1/2} \left(\frac{4\, \text{GeV}}{m_f}\right) \left(\frac{T_{\text{RH}}}{m_\phi}\right)^{1/2} \left(\frac{T_{\text{RH}}}{10\, \text{GeV}}\right)^{1/2},
\tag{A.12}
$$

or equivalently

$$\lambda_{h\phi} \gtrsim 10^{-6}\left(\frac{v_{\text{EW}}}{10\,v_\phi}\right)\left(\frac{g_*}{100}\right)^{1/4}\left(\frac{3}{N_c}\right)^{1/2}\left(\frac{4\,\text{GeV}}{m_f}\right)\left(\frac{T_{\text{RH}}}{m_\phi}\right)^{1/2}\left(\frac{T_{\text{RH}}}{10\,\text{GeV}}\right)^{1/2}. \qquad \text{(A.13)}$$

The exotic Higgs decay $h \to \phi\phi$, has a branching fraction $\text{Br} \approx 10^{-9} \times (\lambda_{h\phi}/10^{-6})^2$ and is safely below collider constraints for $m_\phi$ above the muon threshold (the $m_\phi$ parameter space of interest for our PTs).

## B Initial DM Momentum in the Plasma Frame

We consider the pair production $\phi \to X + X$. Taking the wall to be moving at ultra-relativistic velocity in the positive $z$ direction, the kinematics in the wall frame can be written as

$$\begin{aligned}
p^\phi &= \left(E', 0, 0, -\sqrt{E'^2 - m_\phi^2}\right), \\
p_1^X &= \left(E'[1-x], 0, k_\perp, -\sqrt{E'^2[1-x]^2 - k_\perp^2 - m_{\text{DM}}^2}\right), \\
p_2^X &= \left(E'x, 0, -k_\perp, -\sqrt{E'^2 x^2 - k_\perp^2 - m_{\text{DM}}^2}\right).
\end{aligned} \qquad \text{(B.1)}$$

The pair production probability, in the anti-adiabatic regime, is given by [44]

$$P(\phi \to XX) \simeq \frac{\lambda^2 v_\phi^2}{32\pi^2} \int_0^1 dx\, x(1-x) \int \frac{dk_\perp^2}{(k_\perp^2 + m_{\text{DM}}^2)^2} \simeq \frac{\lambda^2 v_\phi^2}{192\pi^2 m_{\text{DM}}^2}. \qquad \text{(B.2)}$$

From the above, we can also read off the distribution in energy and $k_\perp$ of the outgoing particles.

Azatov et al. also provide a convenient way of calculating the average energy of the outgoing $X$ in the plasma frame. In terms of the incoming energy in the wall frame, $E'$, it is given by

$$\begin{aligned}
\bar{E}_X &= \frac{1}{2}\left[\int_0^1 dx\, x(1-x)\right]^{-1} \\
&\quad \times \left\{\int_0^1 dx\, x(1-x)\gamma_{\text{wp}}\left[E' - \sqrt{E'^2 x^2 - k_\perp^2 - m_{\text{DM}}^2} - \sqrt{E'^2[1-x]^2 - k_\perp^2 - m_{\text{DM}}^2}\right]\right\} \\
&\approx \frac{3\gamma_{\text{wp}} m_{\text{DM}}^2}{2E'}.
\end{aligned} \qquad \text{(B.3)}$$

Here the probability distribution of energy fraction $x$ has been taken into account, and the Lorentz transformation $E = \gamma_{\text{wp}}(E' + v_w p_z')$ has been applied on the sum of the $X$ energies, which also explains the pre-factor $1/2$. In evaluating the integral, the high energy limit been applied $E'x, E'(1-x) \gg m_{\text{DM}}$, and the $k_\perp$ factor has been ignored. This is justified, as the small $x$, large $x$, and large $k_\perp \gtrsim m_{\text{DM}}$ phase spaces are suppressed. Azatov et al. go on to substitute $E' \sim (1 + v_w)\gamma_{\text{wp}} T_n$ to find $\bar{E}_X \sim 3m_{\text{DM}}^2/4T_n$.

Now that we have $\bar{E}_X$ as a function of $E'$, however, we can also take an appropriate average over the incoming flux. First we derive a formula for the $\phi$ flux across the wall. The relative velocity in the $z$ direction between the wall and a particle in the plasma frame with $z$ velocity, $v_z$, is $v_{z,\text{rel}} = v_w - v_z \simeq 1 - \frac{p_z}{E} = 1 - c_\theta$. Here $c_\theta \equiv \cos\theta$ where $\theta$ is the angle between $p_z$ and

the $z$-axis. The flux, $\Phi_\phi = d^2 N_\phi / dA dt$, across the wall in the plasma frame is given by

$$
\begin{aligned}
\Phi_\phi &= \frac{g_\phi}{(2\pi)^3} \int d^3 p f(E) v_{z,\text{rel}} \\
&= \frac{g_\phi}{(2\pi)^2} \int_{-1}^{1} dc_\theta (1 - c_\theta) \int_0^\infty dE E^2 f(E) \\
&= \frac{g_\phi \, \zeta(3) T_n^3}{\pi^2} \, .
\end{aligned}
\tag{B.4}
$$

So $\Phi_\phi$ is just the same as the number density.

Now remembering that $E' = \gamma_{\text{wp}} E(1 - c_\theta)$, the average energy of the $X$ after averaging over the incoming $\phi$ flux is given by

$$
\begin{aligned}
\langle \bar{E}_X \rangle &= \frac{1}{\Phi_\phi} \frac{g}{(2\pi)^3} \int d^3 p f(E) v_{z,\text{rel}} \bar{E}_X \\
&= \frac{g}{\Phi_\phi} \frac{3 m_{\text{DM}}^2}{4\pi^2} \int dE E f(E) \\
&= \frac{\pi^2}{8\zeta(3)} \frac{m_{\text{DM}}^2}{T_n} \simeq \frac{m_{\text{DM}}^2}{T_n} \, .
\end{aligned}
\tag{B.5}
$$

This matches the rough derivation of the Lorentz factor, $\gamma_{\text{xp}} = \langle \bar{E}_X \rangle / m_{\text{DM}} \simeq m_{\text{DM}} / T_n$, given in the main text.

## C  DM Momentum Loss

After the PT, the absolute value of the DM momentum, $p_{\text{DM}}$, evolves with redshift as $p_{\text{DM}} = p_i a_i / a \simeq p_i (t_i / t)^{1/2}$, where the subscript $i$ denotes some initial value, $a$ is the scale factor, and we have assumed $a \propto t^{1/2}$ for consistency with the hypothesis of radiation domination. As a consequence, the rate of momentum loss due to redshift, reads

$$
\left. \frac{dp_{\text{DM}}}{dt} \right|_{\text{redshift}} = \frac{p_{\text{DM}}}{2t} \approx p_{\text{DM}} H \, ,
\tag{C.1}
$$

where in the last equality we have used that the age of the Universe is proportional to Hubble at that time, $t \approx H^{-1}$. Our estimate of $v(t_{\text{eq}})$ is therefore valid provided

$$
\frac{1}{p_{\text{DM}}} \left. \frac{dp_{\text{DM}}}{dt} \right|_{\text{bath}} = \left. \frac{d\log(p_{\text{DM}})}{dt} \right|_{\text{bath}} < H \, ,
\tag{C.2}
$$

where $dp_{\text{DM}} / dt |_{\text{bath}}$ is the rate of momentum loss of a DM particle because of its scatterings with bath particles.

### C.1  Relativistic DM

Consider the DM after the phase transition. As a function of temperature, the DM momentum is given by

$$
p_{\text{DM}} \approx \frac{m_{\text{DM}}^2 T}{T_n T_{\text{RH}}} \, .
\tag{C.3}
$$

In the plasma frame, it is relativistic until $p_{\rm DM} \approx m_{\rm DM}$, i.e. for temperatures

$$T \gtrsim \frac{T_n T_{\rm RH}}{m_{\rm DM}} \tag{C.4}$$

$$\approx 1 \text{ MeV} \left( \frac{10^8 \text{ GeV}}{m_{\rm DM}} \right) \left( \frac{T_{\rm RH}}{10^4 \text{ GeV}} \right) \left( \frac{T_n}{10 \text{ GeV}} \right).$$

Now consider such relativistic DM travelling in the $z$-direction through the plasma frame with energy and $z$-momentum component $E_1 \simeq p_{1z} \equiv p_{\rm DM}$. It undergoes scattering with some particle in the thermal plasma with energy $E_2 \sim T$ (its precise momentum orientation is irrelevant for the following, as $p_{\rm DM} \gg T$, for convenience, we can take it to be in the negative $z$-direction in what follows). We wish to determine the momentum loss rate of the DM in the plasma frame. Denote the initial (final) DM four momentum in the centre-of-mass (COM) frame as $p'_1$ ($p'_3$), and the initial (final) bath particle four momentum in the COM frame as $p'_2$ ($p'_4$). We then have

$$p'_1 = (\sqrt{m_{\rm DM}^2 + p_{\rm CM}^2}, 0, 0, p_{\rm CM}), \tag{C.5}$$

$$p'_2 = (p_{\rm CM}, 0, 0, -p_{\rm CM}), \tag{C.6}$$

$$p'_3 = (\sqrt{m_{\rm DM}^2 + p_{\rm CM}^2}, 0, p_{\rm CM} s_\theta, p_{\rm CM} c_\theta), \tag{C.7}$$

$$p'_4 = (p_{\rm CM}, 0, -p_{\rm CM} s_\theta, -p_{\rm CM} c_\theta), \tag{C.8}$$

where $s_\theta \equiv \sin\theta$, $c_\theta \equiv \cos\theta$, $\theta$ is the usual scattering angle, the COM energy squared is

$$\hat{s} = m_{\rm DM}^2 + 4 p_{\rm DM} T, \tag{C.9}$$

and the COM momentum squared is

$$p_{\rm CM}^2 = \frac{(\hat{s} - m_{\rm DM}^2)^2}{4\hat{s}}. \tag{C.10}$$

For later convenient reference, when we come to find constraints from DM momentum loss, it is useful to denote two temperature regimes according to whether DM is relativistic or not in the COM frame. In the first regime, corresponding to $T \gtrsim \sqrt{T_{\rm RH} T_n}$, we have

$$\hat{s} \approx 4 p_{\rm CM}^2 \approx \frac{4 m_{\rm DM}^2 T^2}{T_n T_{\rm RH}}. \tag{C.11}$$

In the second regime, $T \lesssim \sqrt{T_{\rm RH} T_n}$, and we have

$$\hat{s} \approx m_{\rm DM}^2 \gg 4 p_{\rm CM}^2 \approx \frac{4 m_{\rm DM}^2 T^4}{T_{\rm RH}^2 T_n^2}. \tag{C.12}$$

In both regimes the combination $p_{\rm CM}^2 \hat{s}$, which appears in various expressions below, is approximately the same. It reads

$$p_{\rm CM}^2 \hat{s} \approx \frac{4 m_{\rm DM}^4 T^4}{T_{\rm RH}^2 T_n^2}. \tag{C.13}$$

In any case, to bring the photon energy from the plasma to the COM frame requires a Lorentz boost with

$$\gamma = \frac{p_{\rm CM}}{T + \nu T} \simeq \frac{p_{\rm CM}}{2T}, \tag{C.14}$$

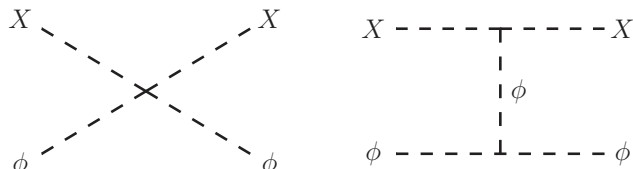

Figure 4: Interactions of the DM with the scalar driving the PT, $\phi$, which can lead to kinetic equilibrium being re-established following the PT.

where we take the relativistic limit $v \simeq 1$. Then, using the Lorentz transformation to boost from the COM frame back into the plasma frame, $E = \gamma(E' + v p'_z)$, we find a momentum loss of the DM in the plasma frame

$$
\begin{aligned}
\delta p_{\mathrm{DM}} &\simeq E_1 - E_3 = \gamma v p_{\mathrm{CM}} (1 - c_\theta) \\
&= -\frac{\gamma v \hat{t}}{2 p_{\mathrm{CM}}} \simeq -\frac{\hat{t}}{4T},
\end{aligned}
\tag{C.15}
$$

where we have used the relation $\hat{t} = -2 p_{\mathrm{CM}}^2 (1 - c_\theta)$. Note our expression above depends on a relativistic boost; eventually, at low $T$, we have $v \ll 1$, the boost reaches the Gallilean limit, and there is an additional suppression. For relativistic DM, we therefore estimate

$$
\frac{d \log(p_{\mathrm{DM}})}{dt} \bigg|_{\mathrm{bath}} \approx \frac{n_{\mathrm{bath}} v_{\mathrm{Møl}}}{p_{\mathrm{DM}}} \int_{-4 p_{\mathrm{CM}}^2}^{0} d\hat{t} \frac{d\sigma}{d\hat{t}} \delta p_{\mathrm{DM}}
\tag{C.16}
$$

$$
\approx -\frac{n_{\mathrm{bath}} v_{\mathrm{Møl}}}{4 p_{\mathrm{DM}} T} \int_{-4 p_{\mathrm{CM}}^2}^{0} d\hat{t} \frac{d\sigma}{d\hat{t}} \hat{t}
\tag{C.17}
$$

$$
\approx -\frac{n_{\mathrm{bath}} v_{\mathrm{Møl}}}{2 p_{\mathrm{CM}} \sqrt{\hat{s}}} \int_{-4 p_{\mathrm{CM}}^2}^{0} d\hat{t} \frac{d\sigma}{d\hat{t}} \hat{t},
\tag{C.18}
$$

where $\sigma$ is the cross section for the process $X\psi \to X\psi$ leading to the momentum loss and $v_{\mathrm{Møl}} \simeq 2$ is the relative (Møller) velocity between the DM and bath particles in the plasma frame. The differential cross section is given by

$$
\frac{d\sigma}{d\hat{t}} = \frac{1}{64 \pi p_{\mathrm{CM}}^2 \hat{s}} |\mathcal{M}|^2,
\tag{C.19}
$$

where $\mathcal{M}$ is the usual matrix element. Given our field content, we will be interested in cross-quartic scalar interactions, and diagrams with scalar exchange in the $t$-channel.

Note that, in the massless limit, $d\sigma/d\hat{t} \propto |\mathcal{M}|^2/\hat{s}^2$. And with our field content, we will always have a $1/\hat{s}^2$ suppression in this quantity. This is qualitatively different to examples featuring vector mediated interactions, such as in Møller scattering or its scalar QED analogue, in which $|\mathcal{M}|^2 \propto \hat{s}^2/\hat{t}^2$ type terms lift the suppression, and lead to IR enhancements in the momentum loss through soft gauge boson exchange. This is the key reason why, in the end, our naive approximation of the momentum loss rate via hard scattering, Eq. (11), gives the appropriate, i.e. the strongest constraint. Note, however, that care must be taken for t-channel diagrams when $p_{\mathrm{CM}}^2 < m_{\mathrm{DM}}^2$, in order to check that the suppression is not lifted by the $p_{\mathrm{CM}}^2$ term in the denominator of Eq. (C.19), leading to a rapid momentum loss. This is what we go on to check below. (Of course, in the deep IR, divergences will also be removed by the mass of the mediating particles.)

### C.1.1 Interactions within the BSM sector

**Scattering with the scalar driving the PT** — We first consider scatterings $X + \phi \rightarrow X + \phi$, for which the Feynman diagrams are shown in Fig. 4. We begin with the cross quartic interaction in Eq. (6). Ignoring interference with the second diagram for now, we have

$$\frac{d\sigma}{d\hat{t}} = \frac{\lambda^2}{64\pi p_{\mathrm{CM}}^2 \hat{s}}. \tag{C.20}$$

Using Eq. (C.18) and demanding Eq. (C.2) hold, we obtain the condition

$$\frac{d\log(p_{\mathrm{DM}})}{dt} \approx n_\phi \frac{\lambda^2 p_{\mathrm{CM}}}{8\pi \hat{s}^{3/2}} \approx \frac{n_\phi \lambda^2}{16\pi \hat{s}} < H. \tag{C.21}$$

In the second approximation, we have used $p_{\mathrm{CM}} \simeq \sqrt{\hat{s}}/2 \gg m_{\mathrm{DM}}$, valid here because we only have to consider high temperatures, as $n_\phi$ becomes Boltzmann suppressed at $T < m_\phi \sim T_{\mathrm{RH}}$. Note Eq. (C.21) is just the same as Eq. (11) in the main text, thus confirming the latter as a suitable estimate. Finally, plugging Eq. (C.11) into Eq. (C.21) and taking $T \simeq T_{\mathrm{RH}}$, we obtain the upper bound on the coupling

$$\lambda < 1.5 \times \left(\frac{m_{\mathrm{DM}}}{10^8 \text{ GeV}}\right)\left(\frac{10 \text{ GeV}}{T_n}\right)^{1/2}. \tag{C.22}$$

It is also interesting to consider a more general case with $m_\phi \lesssim T_{\mathrm{RH}}$ or even $m_\phi \ll T_{\mathrm{RH}}$. Then $n_\phi$ does not become Boltzmann suppressed until lower temperatures. Nevertheless, retaining the temperature dependencies of $\hat{s}$ and $p_{\mathrm{CM}}$ from Eqs. (C.11) and (C.12), we find the strongest constraint on $\lambda$ comes from around $T \approx \sqrt{T_n T_{\mathrm{RH}}}$, when DM turns non-relativistic in the COM frame. The constraint then reads

$$\lambda < 1.5 \times \left(\frac{m_{\mathrm{DM}}}{10^8 \text{ GeV}}\right)\left(\frac{10 \text{ GeV}}{T_n}\right)^{1/4}\left(\frac{10 \text{ GeV}}{T_{\mathrm{RH}}}\right)^{1/4}. \tag{C.23}$$

This collapses to Eq. (C.22) for $T_n = T_{\mathrm{RH}}$ and becomes stronger for $T_n < T_{\mathrm{RH}}$. Thus the allowed parameter space for the $T_n < T_{\mathrm{infl}}$ case becomes somewhat smaller, but still allows for NCDM (for further details see App. D [Fig. 8]).

Realistically, $\phi$ will also have a quartic coupling, $\lambda_\phi$, which will lead to an additional Feynman diagram for the process $X + \phi \rightarrow X + \phi$. This involves t-channel $\phi$ exchange, and so the scattering rate can have an IR enhancement. First, we ignore the interference term and consider only the amplitude squared of the t-channel diagram. We get

$$\frac{d\sigma}{d\hat{t}} = \frac{9\lambda^2 \lambda_\phi^2 v_\phi^4}{16\pi p_{\mathrm{CM}}^2 \hat{s}(\hat{t} - m_\phi^2)^2}. \tag{C.24}$$

This gives a momentum loss

$$\frac{d\log(p_{\mathrm{DM}})}{dt} \simeq n_\phi \frac{9\lambda^2 \lambda_\phi^2 v_\phi^4}{16\pi p_{\mathrm{CM}}^3 \hat{s}^{3/2}} \times \left(\log\left[1 + \frac{4p_{\mathrm{CM}}^2}{m_\phi^2}\right] - \frac{4p_{\mathrm{CM}}^2}{m_\phi^2 + 4p_{\mathrm{CM}}^2}\right). \tag{C.25}$$

We find that Eq. (C.25) gives a much weaker constraint than Eq. (C.21),[3]

$$\lambda < \frac{10^8}{\lambda_\phi}\left(\frac{m_{\mathrm{DM}}}{10^8 \text{ GeV}}\right)^3\left(\frac{10^5 \text{ GeV}}{v_\phi^2}\right)^2\left(\frac{m_\phi}{10^4 \text{ GeV}}\right)^{5/2}\left(\frac{10^4 \text{ GeV}}{T_{\mathrm{RH}}}\right)^{3/2}\left(\frac{10 \text{ GeV}}{T_n}\right)^{3/2}, \tag{C.26}$$

---

[3]We report the numerical constraints on the couplings even when these are nominally $\gg 1$ and thus outside of the realistic perturbative regime. These can then simply be interpreted as meaning that any sensible perturbative choice of the coupling will not lead to issues with DM momentum loss via said process.

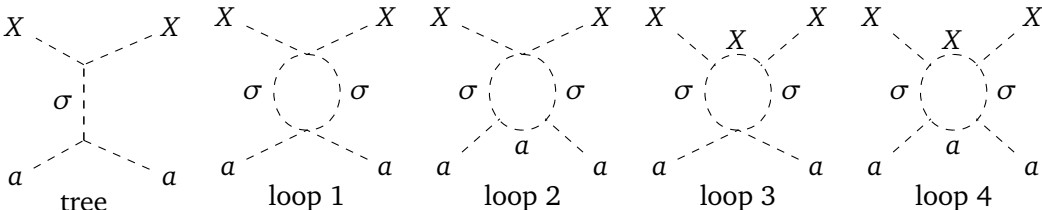

Figure 5: Tree and loop diagrams contributing to the elastic scattering of scalar dark matter $X$ with angular mode $a$.

where we have again allowed for the possibility $m_\phi < T_{\text{RH}}$.

So far we have ignored the interference term. But this cannot realistically give us a stronger limit, as $|2\text{Re}[\mathcal{M}_1\mathcal{M}_2^\dagger]| \leq 2|\mathcal{M}_1||\mathcal{M}_2| \leq |\mathcal{M}_1|^2 + |\mathcal{M}_2|^2$. Indeed, direct computation shows the leading interference term gives a momentum loss which is suppressed by an additional power of $v_\phi^2/\hat{s} \ll 1$, compared to Eq. (C.21) (and also, of course, by the possible Boltzmann factor at $T \lesssim m_\phi \sim T_{\text{RH}}$).

**Scattering with the eventual angular mode.** — In the case where the scalar field driving the phase transition is complex, $\Phi = (v_\phi + \sigma)e^{ia/v_\phi}/\sqrt{2}$, the Goldstone boson $a$ when not eaten by a gauge boson can eventually slow DM down. Scattering of scalar DM with Goldstone bosons are induced by the terms

$$\mathcal{L} \supset \partial_\mu \Phi^\dagger \partial^\mu \Phi - \frac{\lambda}{2}|\Phi|^2 X^2$$
$$\supset \left(\frac{\sigma}{v_\phi} + \frac{\sigma^2}{2v_\phi^2}\right)\partial_\mu a \partial^\mu a - \frac{\lambda}{2}\left(v_\phi \sigma + \frac{\sigma^2}{2}\right)X^2, \tag{C.27}$$

where $\lambda$ is the $X - \phi$ quartic coupling. The matrix element for $Xa \to Xa$, from tree-level $t$-channel exchanges of a radial mode $\sigma$, see Fig. 5 left, reads

$$\mathcal{M} = \lambda \frac{\hat{t}/2 - m_a^2}{\hat{t} - m_\sigma^2}, \tag{C.28}$$

where $m_\sigma$ and $m_a$ are the masses of the radial and angular modes.[4] In the high momentum exchange limit $\hat{t} \gg m_\sigma^2, m_a^2$, we obtain the momentum loss for $Xa \to Xa$

$$\frac{d\log(p_{\text{DM}})}{dt} \simeq n_a \frac{\lambda^2 p_{\text{CM}}}{32\pi \hat{s}^{3/2}}. \tag{C.29}$$

The maximal momentum loss rate is obtained for $T = \sqrt{T_n T_{\text{RH}}}$ when the temperature dependence of the squared energy $\hat{s}$ and DM momentum $p_{\text{CM}}$ in the COM frame changes from $\propto T$ to $\propto T^0$, and from $\propto \sqrt{T}$ to $\propto T$, respectively. We obtain the condition

$$\lambda < \left(\frac{m_{\text{DM}}}{10^8 \text{ GeV}}\right)\left(\frac{10 \text{ GeV}}{T_n}\right)^{1/2}\left(\frac{10 T_n}{T_{\text{RH}}}\right)^{1/4}, \tag{C.30}$$

which becomes competitive with Eq. (C.22) for $T_{\text{RH}} \gtrsim 10 T_n$. In Fig. 8, with orange shading we show the region where momentum loss of DM due to scattering with the Goldstone mode is important. We also study the effects of loop-induced 4-scalar terms. The four diagrams are

---

[4]For scatterings $X + \sigma \to X + \sigma$, we can simply use the computations done for the real scalar $\phi$, and replace $m_\phi \to m_\sigma$.

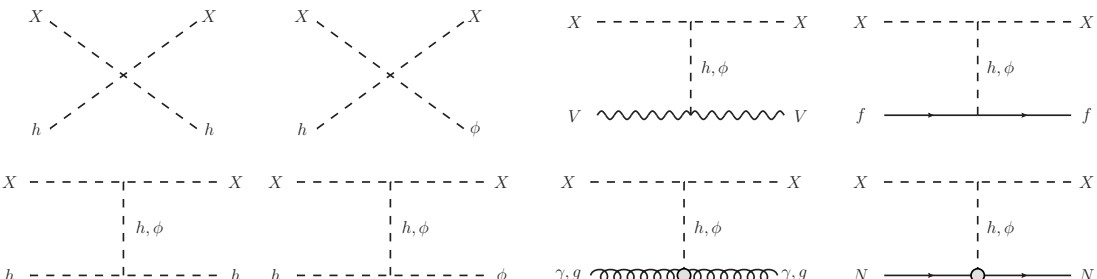

Figure 6: Interactions of the DM with the particles of SM plasma; the SM Higgs, $h$, elementary fermions, $f$, massive gauge bosons, $V$, photons, $\gamma$, gluons, $g$, and nucleons $N$, which can lead to kinetic equilibrium being re-established following the PT. Note at high enough momentum, the DM may instead interact with the partons inside the nucleon, and also break apart the initial nucleon. All corresponding amplitudes of diagrams shown here are suppressed by at least one power of the portal coupling $\lambda_{h\phi}$.

pictured in Fig. 5. The corresponding matrix elements in the large momentum transfer limit $\hat{t} \to \infty$ reads

$$\mathcal{M}_1 + \mathcal{M}_2 \simeq \frac{\lambda}{32\pi^2} \frac{m_\phi^2}{v_\phi^2} \log\left[\frac{m_\phi^2}{-\hat{t}}\right], \tag{C.31}$$

$$\mathcal{M}_3 \simeq \frac{\lambda^2}{128\pi^2} \log^2\left[\frac{m_{\mathrm{DM}}^2}{-\hat{t}}\right], \tag{C.32}$$

$$\mathcal{M}_4 \simeq -\frac{\lambda^2}{32\pi^2}\left(1 + \log^2\left[-\frac{m_{\mathrm{DM}}^4}{\hat{s}\hat{t}}\right]\right). \tag{C.33}$$

We deduce the resulting contribution to the DM momentum loss

$$\frac{d\log(p_{\mathrm{DM}})}{dt} \simeq 10^{-6} \, n_a \frac{\lambda^4 p_{\mathrm{CM}}}{\hat{s}^{3/2}}, \tag{C.34}$$

where we have kept only the contribution from the fourth diagram, since $\lambda > m_\phi^2/v_\phi^2$ in the parameter space of our interest. We obtain the same parametric as for the t-channel in Eq. (C.29) with an additional suppression due to the extra loop factor $(\lambda/16\pi^2)^2$. We conclude that the loop contributions can be neglected.

### C.1.2 Interactions with the SM sector

We now turn to momentum loss due to interactions with the SM bath. These exist once a portal interaction is introduced, to allow $\phi$ to decay rapidly into the SM following the PT, as discussed in App. A. The applicable Feynman diagrams are shown in Fig. 6. These processes are all suppressed $\propto \lambda_{h\phi}^2$, either directly, or through the mixing angle between the two scalars. However, the initial state bath particles may have a different mass threshold, i.e. below $m_\phi \sim T_{\mathrm{RH}}$. We therefore have to check whether the scattering at lower $T$ can lift the $\lambda_{h\phi}^2$ suppression at any points in parameter space, and therefore lead to a more stringent bound from kinetic equilibrium. We shall find that this is not the case, but still provide an overview, for completeness, of the scattering rates below.

**Scattering with the EW Higgs.** — We begin by emphasising that for $m_\phi \lesssim v_{\mathrm{EW}}$, these scattering are necessarily suppressed compared to the scatterings with the $\phi$ we have considered earlier, so we will always be assuming we are in the heavy $m_\phi$ regime for the purposes of the

checks performed here. Compared to Fig. 2, this corresponds to regions outside the current non-cold DM limit, but the discussion could be relevant if the limit is substantially improved. From now we suppress numerical pre-factors from our estimates of the cross sections.

We first consider the quartic interaction leading to inelastic scattering $X + h \to X + \phi$. The differential cross section is given by

$$\frac{d\sigma}{d\hat{t}} \sim \frac{\theta_{h\phi}^2 \lambda^2}{p_{\text{CM}}^2 \hat{s}}, \tag{C.35}$$

which leads to a momentum loss

$$\frac{d\log(p_{\text{DM}})}{dt} \simeq n_h \frac{\theta_{h\phi}^2 \lambda^2 p_{\text{CM}}}{\hat{s}^{3/2}} < H. \tag{C.36}$$

Taking into account the Boltzmann suppression of $n_h$ at temperatures below $T_{\text{EW}}$ and using Eqs. (C.11) and (C.12), one can readily show that a sufficient condition for the above to be below the Hubble rate is

$$\theta_{h\phi} \lesssim \frac{10^{-2}}{\lambda} \left(\frac{m_{\text{DM}}}{10^8 \text{ GeV}}\right) \left(\frac{10^4 \text{ GeV}}{T_{\text{RH}}}\right)^{1/4} \left(\frac{10 \text{ GeV}}{T_n}\right)^{1/4}, \tag{C.37}$$

in both $p_{\text{CM}} \simeq \sqrt{\hat{s}}/2$ and $p_{\text{CM}} \lesssim \sqrt{\hat{s}}/2$ regimes. The condition (C.37) is compatible with our previous constraint (A.9), showing both rapid decay and absence of kinetic equilibrium can be satisfied. Note the example values we have substituted, correspond to an aggressive choice along the kinetic equilibrium line coming from Eq. (11) in Fig. 2, smaller choices of $T_n$ and $T_{\text{RH}}$ would lead to a weaker constraint.

We now move on to consider the t-channel scattering involving external scalar states $X + h \to X + h$. First consider the $\phi$ exchange diagram. The cross section behaves as

$$\frac{d\sigma}{d\hat{t}} \sim \frac{\lambda_{\phi h}^2 \lambda^2 v_\phi^4}{p_{\text{CM}}^2 \hat{s}(\hat{t} - m_\phi^2)^2}. \tag{C.38}$$

Hence, we have an approximate momentum loss

$$\frac{d\log(p_{\text{DM}})}{dt} \sim \frac{n_h \lambda_{\phi h}^2 \lambda^2 v_\phi^4}{p_{\text{CM}}^3 \hat{s}^{3/2}} \log\left[\frac{4p_{\text{CM}}^2}{m_\phi^2}\right] < H. \tag{C.39}$$

This gives a very weak constraint

$$\lambda_{\phi h} \lesssim \frac{10^3}{\lambda} \left(\frac{m_{\text{DM}}}{10^8 \text{ GeV}}\right)^3 \left(\frac{10^5 \text{ GeV}}{v_\phi}\right)^2 \left(\frac{10^4 \text{ GeV}}{T_{\text{RH}}}\right)^{3/2} \left(\frac{10 \text{ GeV}}{T_n}\right)^{3/2}. \tag{C.40}$$

(Here we have assumed parameter space with $p_{\text{CM}} > m_\phi$ at $T \approx m_h$, if instead $p_{\text{CM}} < m_\phi$, the limit would be weakened, due to finite mass propagator effects.) This condition is obviously compatible with our constraint (A.9). Similar or weaker constraints arise for the cross sections coming from the squared amplitudes of the $X + h \to X + h$ via $h$ exchange diagram and the t-channel $X + h \to X + \phi$ diagrams. As argued above, interference terms do not lead to any stronger constraints. We are therefore safe from scatterings with the EW Higgs.

**Scattering with massive EW gauge bosons.** — We now consider scatterings with the massive EW gauge bosons, in order to check whether there can be any additional enhancement compared to the above processes, due to the presence of the external vectors. Taking into account the relative minus sign between the $\phi$ and $h$ exchange diagrams, coming from

the rotation into the mass basis, we find the tree level diagrams squared give a cross sections of the form

$$\frac{d\sigma}{dt} \sim \frac{\lambda_{h\phi}^2 \lambda^2 v_\phi^4}{p_{\text{cm}}^2 \hat{s}} \frac{\left(8m_V^4 + (\hat{t} - 2m_V^2)^2\right)}{(\hat{t} - m_\phi^2)^2 (\hat{t} - m_h^2)^2}, \tag{C.41}$$

where $m_V$ is the gauge boson mass, and we have substituted in for the mixing angle using Eq. (A.7). In the above, we have used the polarization sum completion relation for massive vectors. Note the absence of any dependence on the gauge coupling or EW VEV for the term $\propto \hat{t}^2$ in the numerator. In the high energy limit, this term corresponds to scatterings with longitudinal gauge bosons, which through the Goldstone boson equivalence theorem can be related to scatterings of the DM with would-be EW Goldstone bosons via t-channel $\phi$ exchange. The latter amplitude is manifestly independent of the gauge coupling or EW VEV, which explains the absence of these parameters in the above term of the cross section.

After performing the integral over $\hat{t}$, in the $m_\phi \gtrsim m_V, m_h$ parameter space, we find a momentum loss

$$\frac{d\log(p_{\text{DM}})}{dt} \sim \frac{n_V \lambda_{h\phi}^2 \lambda^2 v_\phi^4}{p_{\text{CM}}^3 \hat{s}^{3/2}} \left( \log\left[1 + \frac{4p_{\text{CM}}^2}{m_\phi^2}\right] - \frac{4p_{\text{CM}}^2}{m_\phi^2 + 4p_{\text{CM}}^2} \right), \tag{C.42}$$

where $n_V$ is the gauge boson number density. The strongest constraint comes from the logarithmic term at temperature $T \approx m_V$. Even for parameter space in which $p_{\text{CM}} > m_\phi$ at such temperatures, the constraint is very weak,

$$\lambda_{h\phi} \lesssim \frac{10^2}{\lambda} \left(\frac{m_{\text{DM}}}{10^8 \text{ GeV}}\right)^3 \left(\frac{10^5 \text{ GeV}}{v_\phi}\right)^2 \left(\frac{10^4 \text{ GeV}}{T_{\text{RH}}}\right)^{3/2} \left(\frac{10 \text{ GeV}}{T_n}\right)^{3/2}, \tag{C.43}$$

again showing compatibility with Eq. (A.9).

**Scattering with fermions.** — Finally we also consider scatterings with the elementary SM fermions. Again taking into account the relative minus sign between the $\phi$ and $h$ exchange diagrams, we find

$$\frac{d\sigma}{d\hat{t}} \sim \frac{\lambda_{h\phi}^2 \lambda^2 v_\phi^4 m_f^2}{p_{\text{CM}}^2 \hat{s}} \frac{(4m_f^2 - \hat{t})}{(\hat{t} - m_\phi^2)^2 (\hat{t} - m_h^2)^2}, \tag{C.44}$$

where $m_f$ is the fermion mass, and we have again substituted in for the mixing angle using Eq. (A.7). In the regime where $m_\phi \gtrsim m_h$, we obtain a momentum loss

$$\frac{d\log(p_{\text{DM}})}{dt} \sim \frac{n_f \lambda_{h\phi}^2 \lambda^2 v_\phi^4 m_f^2}{p_{\text{CM}}^3 \hat{s}^{3/2} m_\phi^2} \frac{4p_{\text{CM}}^2}{4p_{\text{CM}}^2 + m_\phi^2}. \tag{C.45}$$

The strongest constraint comes from momentum loss at $T \approx \text{Max}[T_{\text{RH}}\sqrt{T_n/m_{\text{DM}}}, m_f]$, where the first condition comes from $p_{\text{CM}} > m_\phi$ and the second from simply having an unsuppressed fermion population in the bath. We thus set our hypothetical fermion mass to $m_f = T_{\text{RH}}\sqrt{T_n/m_{\text{DM}}}$, in order to derive the strongest possible constraint (taking into account the actual SM fermion masses would only weaken the derived constraint). With this substitution, together with our earlier assumption $m_\phi \sim T_{\text{RH}}$, we obtain a constraint from demanding the momentum loss be below the Hubble rate,

$$\lambda_{h\phi} \lesssim \frac{10^2}{\lambda} \left(\frac{m_{\text{DM}}}{10^8 \text{ GeV}}\right)^{9/4} \left(\frac{10^5 \text{ GeV}}{v_\phi}\right)^2 \left(\frac{T_{\text{RH}}}{10^4 \text{ GeV}}\right) \left(\frac{10 \text{ GeV}}{T_n}\right)^{3/4}. \tag{C.46}$$

One can readily check that at lower $p_{\text{CM}}$, the momentum loss becomes suppressed by $m_\phi^4$ and eventually also by $m_h^4$, due to the propagators. Thus no stronger constraint arises at lower $p_{\text{CM}}$, even accounting for the possibility of the $m_f^2 > -\hat{t}$ in the numerator of Eq. (C.44). Similar arguments hold if we instead begin with the assumption $m_\phi \lesssim m_h$ (taking into account the lower $T_{\text{RH}}$ this implies for consistency). Comparison of Eq. (C.46) to Eqs. (A.9) and (A.13) shows compatibility with the rapid $\phi$ decay assumption. So we are safe.

**Scattering with photons and gluons.** — The population of photons and gluons does not become Boltzmann suppressed at low $T$ (although the gluons eventually become confined.) We therefore also check whether scatterings of the DM with massless gauge bosons can lead to non-negligible momentum loss at low $T$. Fermionic triangle diagrams in the broken EW phase lead to the effective coupling of the Higgs to gluons via the effective operator

$$\mathcal{L} \sim \frac{\alpha_s}{v_{\text{EW}}} h G^a_{\mu\nu} G^{a\,\mu\nu}. \tag{C.47}$$

Here $\alpha_s$ is the QCD fine structure constant and $G^a_{\mu\nu}$ are the QCD field strength tensors. A similar operator for the photons arises from fermionic triangle diagrams and loop diagrams involving charged gauge bosons,

$$\mathcal{L} \sim \frac{\alpha_{\text{EM}}}{v_{\text{EW}}} h F_{\mu\nu} F^{\mu\nu}, \tag{C.48}$$

where $F_{\mu\nu}$ is the electromagnetic (EM) field strength tensor.

After the $\phi - h$ mixing is taken into account, we find a differential cross section

$$\frac{d\sigma}{d\hat{t}} \sim \frac{\alpha_{\text{EM},s}^2 \lambda^2 \lambda_{h\phi}^2 v_\phi^4}{p_{\text{CM}}^2 \hat{s}} \frac{\hat{t}^2}{(\hat{t}-m_\phi^2)^2(\hat{t}-m_h^2)^2}. \tag{C.49}$$

The contribution to the vertex coming from the top triangle diagram is suppressed at large momentum exchange, $-\hat{t} > m_t^2$, by a factor $\sim m_t^2/(2\hat{t})\log^2(-\hat{t}/m_t^2)$, which in turn suppresses the above cross section for the gluon scattering. The EM counterpart, however, includes effects of the longitudinal W bosons for which — in analogy with the heavy Higgs limit in the decay $h \to \gamma\gamma$ [80] — we do not expect any suppression. Hence, to derive a sufficient constraint in a simple manner, we use the cross section as written above also for $p_{\text{CM}} > m_t$. Considering our benchmark values for which $m_\phi > m_h$, and ignoring the finite $m_h$ for simplicity, we have a momentum loss

$$\frac{d\log(p_{\text{DM}})}{dt} \sim \frac{n_{\gamma,g} \alpha_{\text{EM},s}^2 \lambda^2 \lambda_{h\phi}^2 v_\phi^4}{p_{\text{cm}}^3 \hat{s}^{3/2}} \left( \log\left[ 1 + \frac{4p_{\text{CM}}^2}{m_\phi^2} \right] - \frac{4p_{\text{CM}}^2}{m_\phi^2 + 4p_{\text{CM}}^2} \right), \tag{C.50}$$

where $n_\gamma$ ($n_g$) is the photon (gluon) number density. Note, up to the two powers of the relevant fine structure constant and suppressed loop factors, this is just the same as the scattering with the gauge bosons, Eq. (C.42), which is also dominated by the longitudinal gauge boson contribution at large momentum exchange. For the scattering with the massless gauge bosons, however, we now no longer have the Boltzmann suppression of the bath particles, so the constraint can be somewhat stronger. Using Eq. (C.50), together with Eqs. (C.12) and (C.13), one finds a sufficient condition to avoid momentum loss given by

$$\lambda_{h\phi} \lesssim \frac{10^{-2}}{\alpha_{\text{EM},s}\lambda} \left( \frac{m_{\text{DM}}}{10^8\,\text{GeV}} \right)^{7/4} \left( \frac{10^5\,\text{GeV}}{v_\phi} \right)^2 \left( \frac{T_{\text{RH}}}{10^4\,\text{GeV}} \right) \left( \frac{10\,\text{GeV}}{T_n} \right)^{1/4}, \tag{C.51}$$

where we have used that the strongest constraint comes from when $p_{\text{CM}} \approx m_\phi$ (note for the benchmark values this occurs before the QCD phase transition when the gluons confine). Similarly weak constraints arise for areas of parameter space where we instead have $m_h \gtrsim m_\phi \approx T_{\text{RH}}$. So we are safe.

**Scattering with nucleons.** — After QCD confinement, at $T_{\text{QCD}} \approx 0.1$ GeV, relativistic DM can interact with the nucleons. The latter are non-relativistic as $m_N/T_{\text{QCD}} > 3$. In the plasma frame we write the four-momenta as

$$p_{\text{DM}}^{\mu} \equiv p_1 \simeq (\sqrt{p_{\text{DM}}^2 + m_{\text{DM}}^2}, 0, 0, p_{\text{DM}}), \tag{C.52}$$

$$p_{\text{N}}^{\mu} \equiv p_2 \simeq (m_N, 0, 0, 0). \tag{C.53}$$

The centre-of-mass energy squared is

$$\hat{s} \simeq m_{\text{DM}}^2 + 2p_{\text{DM}}m_N \simeq m_{\text{DM}}^2, \tag{C.54}$$

as $p_{\text{DM}}m_N < m_{\text{DM}}^2$ for $T < T_n T_{\text{RH}}/m_N$ which is always satisfied in our model for $T \leq T_{\text{QCD}}$. The momentum in the centre-of-mass frame is

$$p_{\text{CM}}^2 = \frac{(\hat{s} - (m_{\text{DM}} + m_N)^2)(\hat{s} - (m_{\text{DM}} - m_N)^2)}{4\hat{s}}$$
$$\simeq \left(\frac{p_{\text{DM}}m_N}{m_{\text{DM}}}\right)^2 \simeq \left(\frac{m_{\text{DM}}m_N T}{T_n T_{\text{RH}}}\right)^2. \tag{C.55}$$

Now if $p_{\text{CM}} > m_N$ the incoming DM probes the internal constituents of the nucleon and deep inelastic scattering (DIS) is possible. This translates into a condition $p_{\text{DM}} > m_{\text{DM}}$, so DIS is possible as long as DM is relativistic. Consider now the interaction of the DM with a parton carrying fractional momentum $p'_p = xp_{\text{CM}}$, where $0 < x < 1$. In the DM-nucleon centre-of-mass frame we have four-momenta of the DM and parton

$$p'_{\text{DM}} \equiv p'_1 \simeq (\sqrt{p_{\text{CM}}^2 + m_{\text{DM}}^2}, 0, 0, p_{\text{CM}}), \tag{C.56}$$

$$p'_p \equiv p'_2 \simeq (xp_{\text{CM}}, 0, 0, -xp_{\text{CM}}). \tag{C.57}$$

We now go into the DM-parton COM frame, in which quantities will be denoted with a double prime. Accordingly, the momenta are

$$p''_1 \simeq (\sqrt{p_{\text{CM}}^2 + m_{\text{DM}}^2}, 0, 0, p_{\text{CM}}), \tag{C.58}$$

$$p''_2 \simeq (xp_{\text{CM}}, 0, 0, -xp_{\text{CM}}). \tag{C.59}$$

The COM energy squared is

$$\hat{s}'' = (p'_1 + p'_2)^2$$
$$\simeq m_{\text{DM}}^2 + 2xp_{\text{CM}}m_{\text{DM}}$$
$$\simeq m_{\text{DM}}^2 + 2xp_{\text{DM}}m_N, \tag{C.60}$$

where we have used that $p_{\text{CM}} \ll m_{\text{DM}}$ in our temperature/parameter range of interest.[5] This implies the boost from the prime to the doubly primed frame is a non-relativistic one. From this we find

$$p''_{\text{CM}} \simeq x\frac{p_{\text{DM}}m_N}{m_{\text{DM}}} \simeq xp_{\text{CM}}. \tag{C.61}$$

We consider elastic scatterings at the parton level

$$p''_3 = (\sqrt{m_{\text{DM}}^2 + p''^2_{\text{CM}}}, 0, p''_{\text{CM}}s_\theta, p''_{\text{CM}}c_\theta), \tag{C.62}$$

$$p''_4 = (p''_{\text{CM}}, 0, -p''_{\text{CM}}s_\theta, -p''_{\text{CM}}c_\theta). \tag{C.63}$$

---

[5]The kinematics is thus different to the usual terrestrial DIS, because in terrestrial experiments with electron beams one has $p_{\text{CM}} > m_N > m_e$. In contrast, we have DM playing the role of the electron projectile, and the hierarchy is instead $m_{\text{DM}} > p_{\text{CM}} > m_N$.

The Mandelstam variable at the parton level is

$$\hat{t}'' = 2m_{\text{DM}}^2 - 2p_1'' \cdot p_3'' = -2p_{\text{CM}}''(1-c_\theta). \tag{C.64}$$

The momentum loss of the relativistic DM in the plasma frame, can be estimated from the difference in its energy before/after scattering in said frame,

$$\delta p_{\text{DM}} \simeq \gamma v p_{\text{CM}}''(1-c_\theta) = -\frac{\hat{t}''}{2xm_N}, \tag{C.65}$$

where $\gamma = p_{\text{CM}}/m_N$ is the Lorentz factor of the boost from the plasma to the DM-parton COM frame (at our level of approximation equal to the Lorentz factor for the boost from the plasma to the DM-nucleon COM frame), and $v \simeq 1$ is the associated velocity.

The approximate momentum loss of the DM is therefore

$$\frac{d\log(p_{\text{DM}})}{dt} \tag{C.66}$$

$$\approx \frac{n_N v_{\text{Møl}}}{p_{\text{DM}}} \int_0^1 f_p(x) \left( \int_{-4x^2 p_{\text{CM}}^2}^0 \frac{d\sigma}{d\hat{t}''} \delta p_{\text{DM}} d\hat{t}'' \right) dx$$

$$\approx \frac{-n_N v_{\text{Møl}}}{2p_{\text{DM}} m_N} \int_0^1 \frac{f_p(x)}{x} \left( \int_{-4x^2 p_{\text{CM}}^2}^0 \frac{d\sigma}{d\hat{t}''} \hat{t}'' d\hat{t}'' \right) dx$$

$$\approx \frac{-n_N v_{\text{Møl}}}{2p_{\text{CM}} \sqrt{\hat{s}}} \int_0^1 \frac{f_p(x)}{x} \left( \int_{-4x^2 p_{\text{CM}}^2}^0 \frac{d\sigma}{d\hat{t}''} \hat{t}'' d\hat{t}'' \right) dx,$$

where $f_p(x)$ is the parton distribution function for parton $p$, $v_{\text{Møl}} \simeq 1$, and

$$n_N \sim \text{Max}\left[ (m_N T)^{3/2} e^{-m_N/T}, Y_B T^3 \right], \tag{C.67}$$

is the nucleon density (which is set by the baryon asymmetry, $Y_B \approx 10^{-10}$, at late times). It is also useful to remember the relation,

$$\int_0^1 \sum_p x f_p(x) dx = 1, \tag{C.68}$$

coming from the physical requirement that the sum over the partonic momenta should equal the total nucleon momentum. This implies $\int_0^1 x f_p(x) \le 1$, which we shall use below.

We now consider DM interacting with a quark in the nucleon. We take the cross section from Eq. (C.44), with Mandelstam and momenta in the DM-parton COM frame. We substitute this into Eq. (C.66) and find, assuming $m_h > m_\phi > m_f$, a momentum loss

$$\frac{d\log(p_{\text{DM}})}{dt} \sim \frac{n_N \lambda_{h\phi}^2 \lambda^2 v_\phi^4 m_f^2}{p_{\text{CM}}^3 \hat{s}^{3/2}} \int_0^1 dx \frac{f_f(x)}{x^3} \int_{-4x^2 p_{\text{CM}}^2}^0 d\hat{t}'' \frac{(\hat{t}'' - 4m_f^2)\hat{t}''}{(\hat{t}'' - m_\phi^2)^2 (\hat{t}'' - m_h^2)^2}$$

$$\sim \frac{n_N \lambda_{h\phi}^2 \lambda^2 v_\phi^4 m_f^2}{\hat{s}^{3/2}} \left\{ \int_0^{\frac{m_f}{p_{\text{CM}}}} dx [x f_f(x)] \frac{m_f^2 p_{\text{CM}}}{m_h^4 m_\phi^4} + \int_{\frac{m_f}{p_{\text{CM}}}}^{\frac{m_\phi}{p_{\text{CM}}}} dx [x f_f(x)] \frac{x^2 p_{\text{CM}}^3}{m_h^4 m_\phi^4} \right.$$

$$\left. + \int_{\frac{m_\phi}{p_{\text{CM}}}}^{\frac{m_h}{p_{\text{CM}}}} dx [x f_f(x)] \frac{1}{x^2 m_h^4 p_{\text{CM}}} + \int_{\frac{m_h}{p_{\text{CM}}}}^1 dx [x f_f(x)] \frac{1}{x^6 p_{\text{CM}}^5} \right\}$$

$$\lesssim \frac{n_N \lambda_{h\phi}^2 \lambda^2 v_\phi^4 m_f^2 p_{\text{CM}}}{\hat{s}^{3/2} m_h^4 m_\phi^2}. \tag{C.69}$$

In deriving the above inequality, we have used the following trick: (i) we split the integral over $x$ into effective regions in which the momentum exchange falls above/below relevant mass thresholds, (ii) we then chose $xf_f(x)$ to be a delta function which maximizes each individual contribution. The true contribution is necessarily below this due to the condition (C.68). Thus we avoid having to explicitly substitute in the parton distribution functions and arrive at the bound in the final line. Then demanding our upper bound on the momentum loss be below the Hubble rate, we find a sufficient condition on the coupling

$$\lambda_{h\phi} \lesssim \frac{10^2}{\lambda}\left(\frac{\text{GeV}}{m_f}\right)\left(\frac{m_{\text{DM}}}{10^8\,\text{GeV}}\right)\left(\frac{m_\phi}{10\,\text{GeV}}\right)\left(\frac{T_n}{10\,\text{GeV}}\right)^{1/2}\left(\frac{T_{\text{RH}}}{10\,\text{GeV}}\right)^{1/2}\left(\frac{10^2\,\text{GeV}}{v_\phi}\right)^2\,, \quad \text{(C.70)}$$

where the constraint comes from highest applicable temperature $T \sim 0.1$ GeV. Note we have chosen a somewhat different benchmark point, which for this process leads to a numerically stricter constraint. Also note the constraints for parameter points with $m_\phi > m_h$ are not stricter than the above. So we are safe.

Let us now consider DM interacting with a gluon in the nucleon. We can take our partonic cross section to be as in Eq. (C.49). Consider again $m_h > m_\phi$, then the momentum loss from DIS is

$$\frac{d\log(p_{\text{DM}})}{dt} \sim \frac{n_N \alpha_s^2 \lambda^2 \lambda_{h\phi}^2 v_\phi^4}{\hat{s}^{3/2}}\left\{\int_0^{\frac{m_\phi}{p_{\text{CM}}}} dx[xf_f(x)]\frac{x^4 p_{\text{CM}}^5}{m_h^4 m_\phi^4} + \int_{\frac{m_\phi}{p_{\text{CM}}}}^{\frac{m_h}{p_{\text{CM}}}} dx[xf_f(x)]\frac{p_{\text{CM}}}{m_h^4} \right.$$
$$\left. + \int_{\frac{m_h}{p_{\text{CM}}}}^1 dx[xf_f(x)]\frac{1}{x^4 p_{\text{CM}}^3}\log\left(1 + \frac{4x^2 p_{\text{CM}}^2}{m_h^2}\right)\right\}$$
$$\lesssim \frac{n_N \alpha_s^2 \lambda^2 \lambda_{h\phi}^2 v_\phi^4 p_{\text{CM}}}{\hat{s}^{3/2}m_h^4}\,, \quad \text{(C.71)}$$

where we have used a similar trick as above. Thus a sufficient condition for the momentum loss to be below Hubble reads

$$\lambda_{h\phi} \lesssim \frac{10}{\lambda}\left(\frac{m_{\text{DM}}}{10^8\,\text{GeV}}\right)\left(\frac{T_{\text{RH}}}{10\,\text{GeV}}\right)^{1/2}\left(\frac{T_n}{10\,\text{GeV}}\right)^{1/2}\left(\frac{10^2\,\text{GeV}}{v_\phi}\right)^2\,, \quad \text{(C.72)}$$

where the constraint again comes from the highest applicable temperature $T \sim 0.1$ GeV. Similar weak conditions arise for parameter space in which $m_\phi > m_h$, so we are safe. Although we have dealt with nucleons, scatterings with lighter QCD bound states such as pions would in principle also occur. Around the QCD cofinement temperature, their number density is unsuppressed compared the nucleons. However, the constraints we derived above would be very weak even if we artificially set $Y_B = 1$. Hence, we expect also DIS with pions and other light QCD bound states to give only weak limits, and hence no substantial DM momentum loss.

### C.1.3  Direct coupling of the DM with the Higgs

Due to diagrams of the type shown in Fig. 7, after some running under the renormalization group equations (RGEs), we expect also a portal coupling between the DM and the SM Higgs

$$\mathcal{L} \supset -\lambda_{hx}|H|^2 X^2\,, \quad \text{(C.73)}$$

with $\lambda_{hx} \sim \lambda_{h\phi}$ as $\lambda \sim 1$. This interaction does not lead to any more stringent constraint on the model than what has been considered above. For example, with regard to the diagrams controlling t-channel scatterings with the EW gauge bosons and fermions, there is now no cancellation between propagators, but $v_{\text{EW}}$ enters in numerators instead of $v_\phi$. This compensates

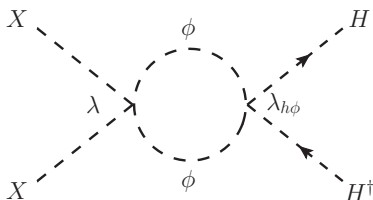

Figure 7: One loop correction contributing to the beta function of the $\lambda_{hx}$ coupling.

and leads to comparable or weaker limits. Let us anyway run through the most important constraints.

**Scattering with the EW Higgs.** — The strongest constraint on scattering with the EW Higgs again comes from the four point vertex. The differential cross section is given by

$$\frac{d\sigma}{d\hat{t}} \sim \frac{\lambda_{hx}^2}{p_{\text{CM}}^2 \hat{s}}. \tag{C.74}$$

The momentum loss is

$$\frac{d\log(p_{\text{DM}})}{dt} \sim \frac{n_h \lambda_{hx}^2 p_{\text{CM}}}{\hat{s}^{3/2}}. \tag{C.75}$$

Assuming $T \gtrsim m_h$, so that the Higgs number density in the plasma is not suppressed, we find a resulting sufficient condition to avoid momentum loss

$$\lambda_{hx} \lesssim 10^{-3} \left(\frac{m_{\text{DM}}}{10^8 \text{ GeV}}\right) \left(\frac{10^4 \text{ GeV}}{T_{\text{RH}}}\right)^{1/4} \left(\frac{10 \text{ GeV}}{T_n}\right)^{1/4}, \tag{C.76}$$

covering both relativistic and non-relativistic DM in the COM frame. The t-channel diagram mediating the DM scattering with $h$ leads to much weaker limits.

**Scattering with massive EW gauge bosons.** — Consider now the scattering $X + V \to X + V$ via a t-channel $h$ propogator. The coupling (C.73) leads to a differential cross section

$$\frac{d\sigma}{d\hat{t}} \sim \frac{\lambda_{hx}^2}{p_{\text{CM}}^2 \hat{s}} \frac{(\hat{t}^2 - 4\hat{t}m_V^2 + 12m_V^4)}{(\hat{t} - m_h^2)^2}. \tag{C.77}$$

The strongest constraint comes from the $\propto \hat{t}^2$ term in the numerator, which from the momentum loss leads to the same limit on $\lambda_{hx}$, Eq. (C.76), as for the scattering with the EW Higgs. The term $\propto m_V^4$ eventually leads to a much weaker constraint. To see this, we calculate the corresponding momentum loss

$$\frac{d\log(p_{\text{DM}})}{dt} \sim \frac{n_V \lambda_{hx}^2 m_V^4}{p_{\text{CM}}^3 \hat{s}^{3/2}} \left(\log\left[1 + \frac{4p_{\text{CM}}^2}{m_h^2}\right] - \frac{4p_{\text{CM}}^2}{m_h^2 + 4p_{\text{CM}}^2}\right). \tag{C.78}$$

Given the cut-off below $T \sim m_V$, together with our parameter space of interest which gives $p_{\text{CM}} > m_h$ at such temperatures, this translates into a very weak constraint

$$\lambda_{hx} \lesssim 10^9 \left(\frac{m_{\text{DM}}}{10^8 \text{ GeV}}\right)^3 \left(\frac{10^4 \text{ GeV}}{T_{\text{RH}}}\right)^{3/2} \left(\frac{10 \text{ GeV}}{T_n}\right)^{3/2}, \tag{C.79}$$

where we have used Eq. (C.13), so we are safe from scattering with the massive EW gauge bosons.

**Scattering with fermions.** — We now consider the $\lambda_{hx}$ induced DM scattering with fermions via t-channel Higgs exchange. We have

$$\frac{d\sigma}{d\hat{t}} \sim \frac{\lambda_{hx}^2 m_f^2}{p_{\text{CM}}^2 \hat{s}} \frac{(4m_f^2 - \hat{t})}{(\hat{t} - m_h^2)^2} \, . \tag{C.80}$$

The momentum loss is dominated by the $\propto \hat{t}$ term in the numerator, because if $-\hat{t}$ is below the EW scale, there is a $m_h^4$ suppression due to the propagator. The momentum loss is given by

$$\frac{d\log(p_{\text{DM}})}{dt} \sim \frac{n_f \lambda_{hx}^2 m_f^2}{p_{\text{CM}}^3 \hat{s}^{3/2}} \left( \frac{16 p_{\text{CM}}^4 + 8 p_{\text{CM}}^2 m_h^2}{4 p_{\text{CM}}^2 + m_h^2} - 2m_h^2 \log\left[1 + \frac{4 p_{\text{CM}}^2}{m_h^2}\right]\right) . \tag{C.81}$$

The strongest constraint occurs for the lowest applicable $T$, and largest applicable $m_f$. This occurs when $\sqrt{T_n T_{\text{RH}}}$ coincides with $m_f$ around the EW scale, but still results in a weak sufficient condition

$$\lambda_{hx} \lesssim 10^3 \left(\frac{m_{\text{DM}}}{10^8 \, \text{GeV}}\right)^2 \left(\frac{10^4 \, \text{GeV}}{T_{\text{RH}}}\right)^{3/4} \left(\frac{10 \, \text{GeV}}{T_n}\right)^{3/4} , \tag{C.82}$$

covering both relativistic and non-relativistic DM in the COM frame. So we are safe from scatterings with fermions.

**Scattering with photons and gluons.** — The differential cross section arising due to the $\lambda_{hx}$ coupling and the effective operator (C.47) or (C.48) is given by

$$\frac{d\sigma}{d\hat{t}} \sim \frac{\alpha_{\text{EM},s}^2 \lambda_{hx}^2}{p_{\text{CM}}^2 \hat{s}} \frac{\hat{t}^2}{(\hat{t} - m_h^2)^2} \, . \tag{C.83}$$

The strongest constraints come from the regime $p_{\text{CM}} > m_h$, where the momentum loss is given by

$$\frac{d\log(p_{\text{DM}})}{dt} \sim \frac{n_{\gamma,g} \alpha_{\text{EM},s}^2 \lambda_{hx}^2 p_{\text{CM}}}{\hat{s}^{3/2}} \, . \tag{C.84}$$

From this, one readily finds a sufficient condition to avoid momentum loss

$$\lambda_{hx} \lesssim \frac{10^{-3}}{\alpha_{\text{EM},s}} \left(\frac{m_{\text{DM}}}{10^8 \, \text{GeV}}\right) \left(\frac{10^4 \, \text{GeV}}{T_{\text{RH}}}\right)^{1/4} \left(\frac{10 \, \text{GeV}}{T_n}\right)^{1/4} , \tag{C.85}$$

encompassing both the $p_{\text{CM}} > m_{\text{DM}}$ and $p_{\text{CM}} < m_{\text{DM}}$ regimes. So we are safe from scatterings with gluons and photons.

**Scattering with nucleons.** — We can also find constraints on $\lambda_{hx}$ from the DIS scatterings with nucleons. The general formula for the momentum loss in this case has of course already been derived above. First we considering DIS scatterings with quarks inside the nucleons. Adapting our cross section in Eq. (C.80) to the DM-parton COM frame, and using Eq. (C.66) we find a momentum loss

$$\frac{d\log(p_{\text{DM}})}{dt} \sim \frac{n_N \lambda_{hx}^2 m_f^2}{\hat{s}^{3/2}} \left\{ \int_0^{\frac{m_f}{p_{\text{CM}}}} dx [x f_f(x)] \frac{m_f^2 p_{\text{CM}}}{m_h^4} + \int_{\frac{m_f}{p_{\text{CM}}}}^{\frac{m_h}{p_{\text{CM}}}} dx [x f_f(x)] \frac{x^2 p_{\text{CM}}^3}{m_h^4} \right.$$
$$\left. + \int_{\frac{m_h}{p_{\text{CM}}}}^1 dx [x f_f(x)] \frac{1}{x^2 p_{\text{CM}}} \right\}$$

$$\lesssim \frac{n_N \lambda_{hx}^2 m_f^2 p_{\text{CM}}}{\hat{s}^{3/2} m_h^2} \, . \tag{C.86}$$

Demanding our upper bound on the momentum loss not exceed the Hubble rate, we find a sufficient condition on the portal coupling

$$\lambda_{hx} \lesssim 10^4 \left(\frac{\text{GeV}}{m_f}\right)\left(\frac{m_{\text{DM}}}{10^8 \,\text{GeV}}\right)\left(\frac{T_n}{10 \,\text{GeV}}\right)^{1/2}\left(\frac{T_{\text{RH}}}{10 \,\text{GeV}}\right)^{1/2}, \tag{C.87}$$

coming from $T \sim 0.1$ GeV. So we are safe.

We can also check the constraint from DIS with the gluons. Adapting the cross section in Eq. (C.83) to the DM-parton COM frame, we find a momentum loss

$$\frac{d\log(p_{\text{DM}})}{dt} \sim \frac{n_N \alpha_s^2 \lambda_{hx}^2}{\hat{s}^{3/2}}\left\{\int_0^{\frac{m_h}{p_{\text{CM}}}} dx[x f_f(x)]\frac{x^4 p_{\text{CM}}^5}{m_h^4} + \int_{\frac{m_h}{p_{\text{CM}}}}^1 dx[x f_f(x)]p_{\text{CM}}\right\}$$

$$\lesssim \frac{n_N \alpha_s^2 \lambda_{hx}^2 p_{\text{CM}}}{\hat{s}^{3/2}}. \tag{C.88}$$

This results in a constraint

$$\lambda_{hx} \lesssim 10 \left(\frac{m_{\text{DM}}}{10^8 \,\text{GeV}}\right)\left(\frac{T_n}{10 \,\text{GeV}}\right)^{1/2}\left(\frac{T_{\text{RH}}}{10 \,\text{GeV}}\right)^{1/2}. \tag{C.89}$$

So we are safe.

## C.2 Non-Relativistic DM

We now turn to considering scatterings when the DM is non-relativistic as measured in the plasma frame. This corresponds to temperatures,

$$T \lesssim \frac{T_n T_{\text{RH}}}{m_{\text{DM}}} \tag{C.90}$$

$$\approx 1 \,\text{MeV}\left(\frac{10^8 \,\text{GeV}}{m_{\text{DM}}}\right)\left(\frac{T_{\text{RH}}}{10^4 \,\text{GeV}}\right)\left(\frac{T_n}{10 \,\text{GeV}}\right).$$

We need to check whether these lead to a stronger constraints on the portal couplings than what we have found above.

Consider first scattering with radiation. In our simplified treatment we take the particles in the plasma frame to have four momenta

$$p_1 = (m_{\text{DM}}, 0, 0, m_{\text{DM}} v_{\text{DM}}), \tag{C.91}$$

$$p_2 = (T, 0, 0, -T). \tag{C.92}$$

Consequently we have

$$\hat{s} \approx m_{\text{DM}}^2 + 2m_{\text{DM}}T(1 + v_{\text{DM}}), \tag{C.93}$$

and

$$p_{\text{CM}} \approx T(1 + v_{\text{DM}}) \approx T. \tag{C.94}$$

To go into the COM frame we must boost with a non-relativistic velocity

$$u = \frac{p_{\text{DM}} - T}{m_{\text{DM}} + T} \simeq \frac{p_{\text{DM}}}{m_{\text{DM}}}. \tag{C.95}$$

In the COM frame, as before, we have four-momenta

$$p_1' = (\sqrt{m_{\text{DM}}^2 + p_{\text{CM}}^2}, 0, 0, p_{\text{CM}}), \tag{C.96}$$

$$p_2' = (p_{\text{CM}}, 0, 0, -p_{\text{CM}}), \tag{C.97}$$

$$p_3' = (\sqrt{m_{\text{DM}}^2 + p_{\text{CM}}^2}, 0, p_{\text{CM}} s_\theta, p_{\text{CM}} c_\theta), \tag{C.98}$$

$$p_4' = (p_{\text{CM}}, 0, -p_{\text{CM}} s_\theta, -p_{\text{CM}} c_\theta). \tag{C.99}$$

Following the scattering, boosting back into the plasma frame via a Gallilean transformation one finds

$$\delta p_{\mathrm{DM}} = -\frac{\hat{t}}{2p_{\mathrm{CM}}}, \tag{C.100}$$

where we have again used $\hat{t} = -2p_{\mathrm{CM}}^2(1-c_\theta)$. We therefore estimate the momentum loss as

$$\left.\frac{d\log(p_{\mathrm{DM}})}{dt}\right|_{\mathrm{bath}} \approx \frac{n_{\mathrm{bath}}v_{\mathrm{M\o l}}}{p_{\mathrm{DM}}}\int_{-4p_{\mathrm{CM}}^2}^0 d\hat{t}\frac{d\sigma}{d\hat{t}}\delta p_{\mathrm{DM}} \tag{C.101}$$

$$\approx -\frac{n_{\mathrm{bath}}v_{\mathrm{M\o l}}}{2p_{\mathrm{DM}}p_{\mathrm{CM}}}\int_{-4p_{\mathrm{CM}}^2}^0 d\hat{t}\frac{d\sigma}{d\hat{t}}\hat{t}. \tag{C.102}$$

(As $p_{\mathrm{CM}} \approx T$ and $v_{\mathrm{M\o l}} \simeq 1$, this is the same as Eq. (C.17) but differs to Eq. (C.18) for relativistic DM.)

**Scattering with photons.** — We can immediately apply our results to non-relativistic DM scattering with photons (for all our parameter space DM is still relativistic at the QCD phase transition). For the $\lambda_{h\phi}$ mediated process we can use the cross section in Eq. (C.49) with appropriate replacement of the strong force associated quantities with their electromagnetic analogues. The momentum loss is then

$$\frac{d\log(p_{\mathrm{DM}})}{dt} \sim \frac{n_\gamma \alpha_{\mathrm{EM}}^2 \lambda^2 \lambda_{h\phi}^2 v_\phi^4 p_{\mathrm{CM}}^5}{p_{\mathrm{DM}}\hat{s}m_h^4 m_\phi^4}. \tag{C.103}$$

Demanding this be below the Hubble rate implies the easily satisfied

$$\lambda_{h\phi} \lesssim \frac{10^{16}}{\lambda}\left(\frac{m_{\mathrm{DM}}}{10^8\,\mathrm{GeV}}\right)^{9/2}\left(\frac{10^5\,\mathrm{GeV}}{v_\phi}\right)^2\left(\frac{10^4\,\mathrm{GeV}}{T_{\mathrm{RH}}}\right)\left(\frac{10\,\mathrm{GeV}}{T_n}\right)^3, \tag{C.104}$$

in the appropriate non-relativistic regime $p_{\mathrm{CM}} \simeq T$. In deriving the above, we have set $T$ to its largest value in the non-relativistic regime, $T \simeq T_n T_{\mathrm{RH}}/m_{\mathrm{DM}}$, and assumed $m_\phi \simeq T_{\mathrm{RH}}$. Because $p_{\mathrm{CM}} \ll m_h^4, m_\phi^4$, the resulting limit is much weaker than for relativistic DM, Eq. (C.51).

The picture repeats. For the $\lambda_{hx}$ mediated process we can use the cross section (C.83) adapted for the photons. The momentum loss is given by

$$\frac{d\log(p_{\mathrm{DM}})}{dt} \sim \frac{n_\gamma \alpha_{\mathrm{EM}}^2 \lambda_{hx}^2 p_{\mathrm{CM}}^5}{p_{\mathrm{DM}}\hat{s}m_h^4}, \tag{C.105}$$

which again gives a very weak constraint

$$\lambda_{hx} \lesssim 10^{18}\left(\frac{m_{\mathrm{DM}}}{10^8\,\mathrm{GeV}}\right)^{9/2}\left(\frac{10^4\,\mathrm{GeV}}{T_{\mathrm{RH}}}\right)^3\left(\frac{10\,\mathrm{GeV}}{T_n}\right)^3. \tag{C.106}$$

Thus we are safe from scatterings with photons.

**Scattering with relativistic fermions.** — We first consider non-relativistic DM scattering with relativistic fermions (for our standard benchmark parameter values the DM turns non-relativistic at $T$ a little above the electron mass). For the $\lambda_{h\phi}$ mediated process we can use the cross section (C.44). The momentum loss is

$$\frac{d\log(p_{\mathrm{DM}})}{dt} \sim \frac{n_f \lambda^2 \lambda_{h\phi}^2 v_\phi^4 m_f^2 p_{\mathrm{CM}}^3}{p_{\mathrm{DM}}\hat{s}m_h^4 m_\phi^4}. \tag{C.107}$$

We can then substitute $m_f = m_e$ as this is the only SM fermion of relevance in this regime. By demanding the momentum loss be below the Hubble rate find

$$\lambda_{h\phi} \lesssim \frac{10^{14}}{\lambda} \left( \frac{m_{\text{DM}}}{10^8 \text{ GeV}} \right)^{7/2} \left( \frac{10^5 \text{ GeV}}{v_\phi} \right)^2 \left( \frac{10 \text{ GeV}}{T_n} \right)^2 , \tag{C.108}$$

where the strongest constraint again comes from $T \simeq T_n T_{\text{RH}}/m_{\text{DM}}$, and we have set $m_\phi \sim T_{\text{RH}}$. For the $\lambda_{hx}$ mediated process we instead use the cross section (C.80) to find the momentum loss,

$$\frac{d\log(p_{\text{DM}})}{dt} \sim \frac{n_f \lambda_{hx}^2 m_f^2 p_{\text{CM}}^3}{p_{\text{DM}} \hat{s} m_h^4} . \tag{C.109}$$

The resulting constraint is

$$\lambda_{hx} \lesssim 10^{16} \left( \frac{m_{\text{DM}}}{10^8 \text{ GeV}} \right)^{7/2} \left( \frac{10^4 \text{ GeV}}{T_{\text{RH}}} \right)^2 \left( \frac{10 \text{ GeV}}{T_n} \right)^2 , \tag{C.110}$$

which does not pose any problems.

**Scattering with non-relativistic fermions.** — We can also consider scattering with non-relativistic fermions. In the current context this means electrons and nucleons. As we shall see below, the low $p_{\text{CM}}$ here means we can take the DM to be effectively interacting with the entire nucleon rather than probing its internal structure. The number density of the fermions is taken approximately as

$$n_f \sim \text{Max} \left[ (m_f T)^{3/2} e^{-m_f/T}, Y_{\text{B}} T^3 \right] , \tag{C.111}$$

where the electron density at low $T$ is approximately related to the baryon asymmetry, $Y_{\text{B}}$, in order for the Universe to be net EM charge neutral. In our simplified treatment we take the particles in the plasma frame to have four momenta

$$p_1 \approx (m_{\text{DM}}, 0, 0, m_{\text{DM}} v_{\text{DM}}), \tag{C.112}$$

$$p_2 = (m_f, 0, 0, 0), \tag{C.113}$$

as the SM fermion momentum is always negligible compared to $p_{\text{DM}}$. Here we have

$$\hat{s} \approx m_{\text{DM}}^2 + m_f^2 + 2 m_{\text{DM}} m_f , \tag{C.114}$$

and

$$p_{\text{CM}} \simeq \frac{m_f}{m_{\text{DM}}} p_{\text{DM}} \simeq \frac{m_{\text{DM}} m_f T}{T_n T_{\text{RH}}} . \tag{C.115}$$

As before we have

$$\delta p_{\text{DM}} = -\frac{\hat{t}}{2 p_{\text{CM}}} , \tag{C.116}$$

and the approximate momentum loss

$$\frac{d\log(p_{\text{DM}})}{dt} \bigg|_{\text{bath}} \approx \frac{n_f v_{\text{Møl}}}{p_{\text{DM}}} \int_{-4 p_{\text{CM}}^2}^{0} d\hat{t} \frac{d\sigma}{d\hat{t}} \delta p_{\text{DM}} \tag{C.117}$$

$$\approx -\frac{n_f v_{\text{Møl}}}{2 p_{\text{DM}} p_{\text{CM}}} \int_{-4 p_{\text{CM}}^2}^{0} d\hat{t} \frac{d\sigma}{d\hat{t}} \hat{t} . \tag{C.118}$$

The relative velocity is approximately

$$v_{\text{Møl}} \sim \text{Max} \left[ \frac{m_{\text{DM}} T}{T_n T_{\text{RH}}}, \sqrt{\frac{T}{m_f}} \right].$$  (C.119)

The first term is simply the speed of the DM in the plasma frame and the second the fermion speed. The latter follows from the usual non-relativsitic relation with the kinetic energy of the fermion, taken to be $\sim T$ (as these are still kinetically coupled to the photon bath). Throughout this regime the electrons are always faster than the DM while the nucleons are slower down to keV scales, even assuming our extremal benchmark parameter point.

Using our previously derived cross sections for the scattering with fermions, we find the momentum loss for the $\lambda_{h\phi}$ dependent scattering,

$$\frac{d\log(p_{\text{DM}})}{dt} \sim \frac{n_f \lambda^2 \lambda_{h\phi}^2 v_\phi^4 y_f^2 m_f^4 p_{\text{CM}} v_{\text{Møl}}}{p_{\text{DM}} \hat{s} m_h^4 m_\phi^4},$$  (C.120)

where $y_f \equiv 1 \ (\sim 0.2)$ for electrons (nucleons) as the latter are composite and one must include the effective Higgs-nucleon coupling [81]. For the $\lambda_{hx}$ dependent scattering we have momentum loss

$$\frac{d\log(p_{\text{DM}})}{dt} \sim \frac{n_f \lambda_{hx}^2 y_f^2 m_f^4 p_{\text{CM}} v_{\text{Møl}}}{p_{\text{DM}} \hat{s} m_h^4}.$$  (C.121)

From these, the strictest bounds on the couplings come from scatterings with the nucleons at the highest temperatures for which DM is non-relativistic in the plasma frame $T \simeq T_n T_{\text{RH}}/m_{\text{DM}}$. They read

$$\lambda_{h\phi} \lesssim \frac{10^{11}}{\lambda} \left( \frac{m_{\text{DM}}}{10^8 \, \text{GeV}} \right)^2 \left( \frac{10^5 \, \text{GeV}}{v_\phi} \right)^2 \left( \frac{m_\phi}{10^4 \, \text{GeV}} \right)^2 \left( \frac{10^4 \, \text{GeV}}{T_{\text{RH}}} \right)^{1/2} \left( \frac{10 \, \text{GeV}}{T_n} \right)^{1/2},$$  (C.122)

and

$$\lambda_{hx} \lesssim \frac{10^{13}}{\lambda} \left( \frac{m_{\text{DM}}}{10^8 \, \text{GeV}} \right)^2 \left( \frac{10^4 \, \text{GeV}}{T_{\text{RH}}} \right)^{1/2} \left( \frac{10 \, \text{GeV}}{T_n} \right)^{1/2}.$$  (C.123)

We are therefore safe.

## C.3 Summary of the DM scattering constraints

We thus conclude our examination of momentum loss. The strongest constraint in general came from hard $X + \phi \to X + \phi$ scatterings at $T \approx m_\phi \sim T_{\text{RH}}$, given after a careful derivation in Eq. (C.21), and given via an approximation in the main paper as Eq. (11). If $\phi$ is complex, scattering with the angular mode must also be accounted for $T_{\text{RH}} \gtrsim 10 T_n$, compare (C.30) and (C.21). The strongest constraints on the portal couplings, in contrast, arose from hard scatterings with the EW Higgs $X + h \to X + \phi$, $X + h \to X + h$ in the regime of relativistic DM in the plasma frame. These limits were given in Eqs. (C.37) and (C.76) and are easily compatible with the couplings required for rapid $\phi$ decay following the PT, given in Eqs. (A.9), (A.10), (A.12), and (A.13).

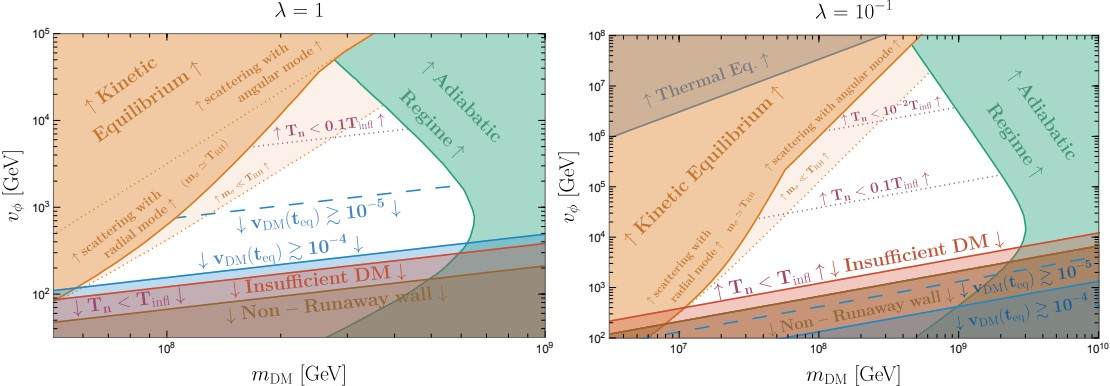

Figure 8: Heavy non-cold DM from fast bubble walls in the plane of $v_\phi$ vs DM mass $m_{\rm DM}$, for $T_n < T_{\rm infl}$. The heavy DM is efficiently produced by fast bubble walls (outside of the green area), compatible with Lyman-$\alpha$ bound (outside of the blue area - here taking $m_{\rm WDM} > 3$ keV), and kinetically decoupled from the bath (outside of the orange area). Bounds from kinetic equilibration via scatterings of DM with light radial and angular modes $\sigma$ and $a$, Eqs. (C.23) and (C.30), have also been indicated. Future 21-cm reach is shown with a dashed blue line. Amount of supercooling is shown with dotted purple lines. At fixed DM abundance, the decrease of the DM-scalar mixing $\lambda$ (from left to right) leads to an increase of the phase transition scale $v_\phi$ which implies a longer redshift of the DM momentum, and results in colder DM. In red, the yield $Y_{\rm DM}$ in Eq. (9) is insufficient to explain DM. In brown, the Bodeker&Moore criterium [37] $\mathcal{P}_{\rm LO} > \Lambda_{\rm vac}$ is satisfied and the acceleration of bubble walls is stopped by thermal friction (we chose $g_a = 20$ and $c_{\rm vac} = 0.01$). In gray (right panel), the reheating temperature is larger than the DM freeze-out temperature $T_{\rm RH} > T_{\rm FO}$ and DM goes back into thermal equilibrium (we assumed the maximal annihilation cross-section allowed by unitarity).

# D   Analytic Derivation of the Coupling and Mass Scales

In the following, it will be useful to remember that the temperature of matter-radiation equality, the DM mass, and yield are related by

$$Y_{\rm DM} m_{\rm DM} = \frac{3}{4} \frac{g_*(T_\gamma^{\rm eq})}{g_{*s}(T_\gamma^{\rm eq})} T_\gamma^{\rm eq} - Y_B m_N \tag{D.1}$$

$$\simeq 0.54\, T_\gamma^{\rm eq} \simeq 0.43~{\rm eV}\,. \tag{D.2}$$

## D.1   The quartic coupling

In order to have NCDM, the coupling $\lambda$ cannot be arbitrarily small. Requiring $v(t_{\rm eq})$, Eq. (10), be above some reference value $v_{\rm lim}$ we find

$$\lambda \gtrsim \left(\frac{v_{\rm lim}}{10^{-4}}\right)^{1/2} \left(\frac{c_{\rm vac}}{10^{-2}}\right)^{1/4} \left(\frac{g_*}{10^2}\right)^{5/12} \left(\frac{T_{\rm RH}}{T_n}\right) \left(\frac{T_{\rm RH}}{T_{\rm infl}}\right), \tag{D.3}$$

where we have used the DM yield, Eq. (9), to relate temperatures and mass scales appearing in the expressions. The temperature ratios appearing above are at their minima, unity, precisely at the vacuum dominated border $T_n = T_{\rm RH} = T_{\rm infl}$, so we immediately get a lower bound on $\lambda$. The coupling $\lambda$ is of course limited from above by the usual arguments from perturbitivity.

## D.2 The DM mass scale

To avoid a return to kinetic equilibrium following the PT, we impose Eq. (11), which gives a lower bound on the DM mass

$$
\begin{aligned}
m_{\text{DM}} > 8.0 \times 10^7 \text{ GeV} \\
\times g_\phi^{2/3} \lambda^{2/3} \left( \frac{c_{\text{vac}}}{10^{-2}} \right)^{1/6} \left( \frac{g_*}{10^2} \right)^{1/6} \left( \frac{M_{\text{Pl}}^{2/3} T_\gamma^{\text{eq}\,1/3}}{1.7 \times 10^9 \text{ GeV}} \right) \left( \frac{T_{\text{RH}}}{T_n} \right)^{1/3} \left( \frac{T_{\text{RH}}}{T_{\text{infl}}} \right)^{2/3}.
\end{aligned} \tag{D.4}
$$

Here we have explicitly included the factor $(M_{\text{Pl}}^2 T_\gamma^{\text{eq}})^{1/3} \simeq 1.7 \times 10^9$ GeV, to show the scaling with these cosmological quantities. The temperature ratios appearing are at least one, and $\lambda$ is bounded from below, so we obtain a lower bound on the DM mass.

Conversely, we can use the anti-adiabaticity condition, coming from Eqs. (5) and (8), to find an upper bound on the DM mass

$$
\begin{aligned}
m_{\text{DM}} < \frac{9.4 \times 10^8 \text{ GeV}}{\lambda^{2/3}} \\
\times \left( \frac{c_{\text{vac}}}{10^{-2}} \right)^{1/3} \left( \frac{30}{A_{\text{bub}} \beta_H} \right)^{2/3} \left( \frac{M_{\text{Pl}}^{2/3} T_\gamma^{\text{eq}\,1/3}}{1.7 \times 10^9 \text{ GeV}} \right) \left( \frac{T_n}{T_{\text{RH}}} \right)^{1/3},
\end{aligned} \tag{D.5}
$$

where the temperature ratio is now at most unity. Thus from Eqs. (D.5), (D.4), (D.3), together with perturbativity of the coupling, we arrive at the DM mass scale $m_{\text{DM}} \sim (0.1 - 1)(M_{\text{Pl}}^2 T_\gamma^{\text{eq}})^{1/3} \sim (10^8 - 10^9)$ GeV.

## D.3 The scale of the VEV

As shown in Fig. 8, fast bubble walls produce NCDM if the VEV of the scalar driving the PT is around the electroweak scale $v_\phi \sim 0.1$ TeV. In order to explain this coincidence, we can also derive this scale analytically. Avoiding kinetic equilibrium gives a lower bound

$$
v_\phi > 110 \text{ GeV} \, \frac{g_\phi^{1/3}}{\lambda^{2/3}} \left( \frac{g_*}{10^2} \right)^{5/12} \left( \frac{c_{\text{vac}}}{10^{-2}} \right)^{1/12} \left( \frac{M_{\text{Pl}}^{1/3} T_\gamma^{\text{eq}\,2/3}}{1.2 \text{ GeV}} \right) \left( \frac{T_{\text{RH}}}{T_n} \right)^{5/3} \left( \frac{T_{\text{RH}}}{T_{\text{infl}}} \right)^{1/3}. \tag{D.6}
$$

The requirement for anti-adiabaticity gives an upper bound

$$
v_\phi < \frac{360 \text{ GeV}}{\lambda^{4/3}} \left( \frac{g_*}{10^2} \right)^{1/2} \left( \frac{c_{\text{vac}}}{10^{-2}} \right)^{1/6} \left( \frac{30}{A_{\text{bub}} \beta_H} \right)^{1/3} \left( \frac{M_{\text{Pl}}^{1/3} T_\gamma^{\text{eq}\,2/3}}{1.2 \text{ GeV}} \right) \left( \frac{T_{\text{RH}}}{T_n} \right)^{4/3}. \tag{D.7}
$$

The range of the VEV in the NCDM region — which is centered around $T_n \sim T_{\text{infl}}$ — is therefore roughly $v_\phi \sim (10^2 - 10^3)(M_{\text{Pl}} T_\gamma^{\text{eq}\,2})^{1/3} \sim (10^2 - 10^3)$ GeV. Indeed, in the case $T_n \geq T_{\text{infl}}$ the range of the VEV can only be tightened by considering some $T_n > T_{\text{infl}}$. While in the case $T_n < T_{\text{infl}}$, we can restrict the range of the VEV by requiring the DM to have some non-negligible $v(t_{\text{eq}}) = v_{\text{lim}}$, which implies a given temperature ratio $T_{\text{RH}}/T_n$. For a lower bound, we thus have

$$
v_\phi > 115 \text{ GeV} \times g_\phi^{1/3} \lambda \left( \frac{10^2}{g_*} \right)^{5/18} \left( \frac{10^{-2}}{c_{\text{vac}}} \right)^{1/3} \left( \frac{10^{-4}}{v_{\text{lim}}} \right)^{5/6} \left( \frac{M_{\text{Pl}}^{1/3} T_\gamma^{\text{eq}\,2/3}}{1.2 \text{ GeV}} \right). \tag{D.8}
$$

And for an upper bound we have

$$
v_\phi < 370 \text{ GeV} \left( \frac{10^2}{g_*} \right)^{1/18} \left( \frac{10^{-2}}{c_{\text{vac}}} \right)^{1/6} \left( \frac{10^{-4}}{v_{\text{lim}}} \right)^{2/3} \left( \frac{30}{A_{\text{bub}} \beta_H} \right)^{1/3} \left( \frac{M_{\text{Pl}}^{1/3} T_\gamma^{\text{eq}\,2/3}}{1.2 \text{ GeV}} \right). \tag{D.9}
$$

Thus conclusively showing that the range of the VEV in the current NCDM region is approximately $v_\phi \sim (10^2 - 10^3)(M_{\text{Pl}} T_\gamma^{\text{eq}\,2})^{1/3} \sim (10^2 - 10^3)$ GeV.

On the other hand, ignoring the requirement of being close to the current NCDM bound, we can find where the bounds (D.6) and (D.7) intersect, in the case $T_n < T_{\text{infl}}$. This gives the absolute upper bound,

$$v_\phi \lesssim \frac{50 \text{ TeV}}{g_\phi^{4/3} \lambda^4} \left( \frac{g_*}{10^2} \right)^{5/6} \left( \frac{c_{\text{vac}}}{10^{-2}} \right)^{1/2} \left( \frac{30}{A_{\text{bub}} \beta_H} \right)^{5/3} \left( \frac{M_{\text{Pl}}^{1/3} T_\gamma^{\text{eq}\,2/3}}{1.2 \text{ GeV}} \right), \qquad \text{(D.10)}$$

which corresponds to the peak of the allowed region in Fig. 8.

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
