# Peer review of "Hot and heavy dark matter from a weak scale phase transition"

_SciPost Physics, doi:SciPost Phys. 14, 033 (2023)_

## Round 1 · Referee Report · Anonymous (Referee 1) · 2022-9-26

Report

This article discusses a very interesting scenario in which the dark matter is heavy and non-cold due to a supercooled phase transition. The authors perform a detailed and careful analysis on this subject, and show this scenario can be probed by future gravitational wave interferometers. I have two minor suggestions. 1) When calculating the gravitational wave, the authors only present one with Tn < Tinflation. I would suggest the authors add another benchmark point, with a different dark matter mass, reheating temperature, and Tn> Tinflation. 2) Can the authors comment briefly on the complementarity between different searches for this scenario? I think those will help the readers to get a more complete picture about this scenario.

---

## Round 1 · Referee Report · Anonymous (Referee 2) · 2022-9-28

Strengths

The authors are interested in the production of super-heavy, non-cold dark matter during a first order cosmological phase transition and the associated signatures. This dark matter production mechanism has been studied over the last few years by other groups. In the work under consideration, the authors emphasize how the dark matter will be produced with a large boost in the early universe, and even at the time of radiation-matter equality, the dark matter may have a non-negligible speed. This implies that the scenario can be constrained by probes of dark matter substructure, such as Lyman-alpha forest observations. Additionally, the authors derive predictions for the stochastic gravitational wave radiation that would be generated at the first order phase transition, and argue that this signal may be within reach of future space-based interferometer experiment LISA.

Weaknesses

1 -- The authors build upon earlier results [40-42], which I haven't studied carefully. I find it remarkable that the dark matter can acquires a Lorentz factor in the plasma frame that's as large as \gamma_{xp} ~ m_{DM} / T_n ~ 10^6 - 10^8. I suppose that the momentum kick comes from the patch of domain wall where the particle collided? I wonder whether this wouldn't have a large back reaction on the dynamics of the wall. If the impacts were sufficiently spread out, would the wall be driven away from a planar configuration? Anyway, this is hardly a shortcoming of the current paper, but rather my own confusion about [40-42].

Report

In my view the manuscript meets the journal's expectations for publication. The analysis is comprehensive and thorough. The idea is novel and provides avenues for future research. The manuscript is well-written and organized clearly. (I particularly appreciate that the authors were able to consolidate many of the details into appendices, so as to allow the main message to come through more clearly in the body of the paper.). I am happy to recommend the article for publication.

Requested changes

1 -- One thought. A recent study ( https://arxiv.org/abs/2203.05750 ) reported a strong limit on ultra-light dark matter from ultra-faint dwarf galaxies. When the constraints are expressed as lower limits on the dark matter mass, the bound is roughly an order of magnitude stronger than existing limits from Lyman-alpha forest observations. Typically these types of measurements leads to limits on both ULDM and WDM (non-negligible velocity at RM equality). Do these observations of UFD galaxies provide meaningful constraints on the model that the authors study?

---

## Round 2 · Referee Report · Anonymous (Referee 2) · 2022-10-24

Report

The authors have addressed the concerns raised in my previous report, and I am now happy to recommend the article for publication in SciPost Physics. By identifying that dark matter produced from fast bubbles leads to large free-streaming lengths, this work opens a new pathway in an existing field of study (dark matter phenomenology) and it also provides a novel / synergetic link between different research areas (dark matter model building & structure formation).

---

## Round 2 · Referee Report · Anonymous (Referee 1) · 2022-11-4

Report

The authors have addressed my concerns in the previous report, and I am happy to recommend it for publication in SciPost.

---

## Round 2 · Author Response

We thank the referees for their reports. We give respond to the points raised by the referees below.

REPLY TO REPORT 1

The referee made a good point that we should have been clearer regarding the gravitational wave (GW) signal in the T_n > T_infl case. We have now included a LISA contour in the summary plot for the T_n > T_infl case, i.e. Fig. 2 left. As can be seen, this is outside the region of validity for our DM parameter space, but should allow the reader to better understand what is going on. We have also included example GW spectra for the T_n > T_infl case in the new Fig. 3 left. The captions of the figures have been suitably adjusted to provide some more information. We have also expanded our discussion at the end of Sections 4 and 5 to qualitatively describe the behaviour of the gravitational wave signal regions and discuss the complementarity of the GW signal and current/future DM free streaming constraints.

REPLY TO REPORT 2

Regarding the backreaction of the pair production on the wall, it is true we have not attempted to solve the scalar equations of motion in the presence of such an effect, in order to calculate any local deformation of the wall etc. On the other hand, as a more modest first check, we can estimate the corrections to the pressure given in Eq. (4), due to the pair creation effect (they were also pointed out in https://arxiv.org/abs/2010.02590). We have now added an explanation regarding both these points at the end of Sec. 3.

The very interesting recent study pointed out by the referee, 2203.05750, uses some distinct features of Fuzzy Dark Matter (FDM) to put constraints on such models from the Segue 1 and Segue 2 Ultra-Faint Dwarf (UFD) galaxies. The analysis exploits the wave interference effects in FDM which leads to dynamical heating of stellar orbits. This effect is not present in warm dark matter (WDM) models. Although the latter do suppress the formation of the low mass halos which would host Milky Way satellite galaxies. This effect is what allows one to place constraints on WDM (and our model) via observations of Milky Way satellites, e.g. as done in 2008.00022 (our Ref. [8]), which in part uses Pan-STARRS1 data including Segue 1 and Segue 2 (see 1912.03302). Although the UFD galaxies are (partly) the same, the heating effect allowed 2203.05750 to place a much more stringent lower bound ~10^-19 eV on FDM from the UFD, compared to the bound of ~10^-21 eV, coming from Halo suppression derived in 2008.00022.

OTHER COMMENTS

Independently, we also uncovered a factor of 1/2 correction to the production probability, Eq. (7), and have updated all plots/equations/appendices as required. In order to display a non-cold DM region in Fig. 2 with the choice lambda = 1, we have changed the fiducial choice of the warm dark matter (WDM) limit from m_wdm = 3 keV to m_wdm = 5 keV, which is supported by a number of the papers given in Refs. [3-14], although some caution is required, see in particular [7]. (Note with a choice of m_wdm = 3 keV, the non-cold DM region in Fig. 2 would sit almost precisely on the T_n = T_infl contour, when setting lambda = 1. Somewhat larger choices of lambda would still allow for the region to appear with m_wdm = 3 keV, but we prefer to keep the coupling as before.)

Finally, we have also added references to 1911.02663, which derives limits on warm dark matter from stellar streams, and 1912.02830, 1912.04238, which present an alternative mechanism in which DM interacts with bubble walls during a phase transition.

---

## Round 2 · List of Changes

The changes to the manuscript have been indicated in red. A list of changes is given below.

  • included the stellar stream limit in the abstract and introduction.
  • changed the benchmark/fiducial limit used from m_wdm = 3 keV to m_wdm = 5 keV.
  • corrected the production probability, Eq. (7), and dependent quantities: Eqs. (9), (30), (167)-(174), Figs. 2, 3, and 8.
  • added discussion at the end of Sec. 3 regarding friction/backreaction on the wall from the pair production.
  • added the LISA contour to Fig. 2 left and adjusted caption.
  • added Fig. 3 left to show the gravitational wave signal in the T_n > T_infl regime.
  • added discussion in Secs. 4 and 5 regarding the signal in the T_n > T_infl regime and complementarity between the gravitational wave and warm dark matter searches.

---

## Editorial Decision

published